# Bridging or Breaking: Impact of Intergroup Interactions on Religious Polarization

**ACM Reference Format:**
Anonymous Author(s). 2023. Bridging or Breaking: Impact of Intergroup Interactions on Religious Polarization. In *Proceedings of ACM Conference (Conference'17)*. ACM, New York, NY, USA, 35 pages. https://doi.org/XXXXXXX.XXXXXXX

## ABSTRACT

While exposure to diverse viewpoints may reduce polarization, it can also have a *backfire effect* and exacerbate polarization when the discussion is adversarial. Here, we examine the question whether intergroup interactions around important events affect polarization between majority and minority groups in social networks. We compile data on the religious identity of nearly 700,000 Indian Twitter users engaging in COVID-19-related discourse during 2020. We introduce a new measure for an individual's group conformity based on contextualized embeddings of tweet text, which helps us assess polarization between religious groups. We then use a meta-learning framework to examine heterogeneous treatment effects of intergroup interactions on an individual's group conformity in the light of communal, political, and socio-economic events. We find that for political and social events, intergroup interactions reduce polarization. This decline is weaker for individuals at the extreme who already exhibit high conformity to their group. In contrast, during communal events, intergroup interactions can increase group conformity. Finally, we decompose the differential effects across religious groups in terms of emotions and topics of discussion. The results show that the dynamics of religious polarization are sensitive to the context and have important implications for understanding the role of intergroup interactions.

## 1 INTRODUCTION

Polarization between identity groups is an important driver of social unrest and may adversely affect a nation's economic growth and responses to crises [15, 16]. However, it is less clear how polarization evolves during crises. On the one hand, collective suffering may foster within-group solidarity [6]. On the other hand, it may lead to attribution of blame on the "outside" group or increase between-group competition for limited resources [24]. These behaviors might be accentuated among people who have a tendency to interact only within their groups, and hence might have a restricted information environment [23]. Polarization has also been widely studied in the social-media context. In particular, social media platforms such as Twitter have become more polarized over time [19]. Consistent with this, political polarization decreased for users who deactivated their Facebook accounts before the US elections [1]. The depolarization however, depends on an individual's background and this pattern might reverse for individuals having homogeneous offline social networks [4]. Interactions within social media platforms might have

varying impacts on polarization. In an unfavourable environment, exposure to outgroup viewpoints may lead to stereotype formation and increase polarization. Bail et al. [5] conduct a field experiment and find that republicans (democrats) who were offered financial incentives to follow a liberal (conservative) twitter bot became more conservative (liberal).

In this paper, we examine the impact of intergroup interactions on polarization on Indian Twitter between religious majority and minority groups in the context of COVID-19 related events. We consider an individual to be engaging in intergroup interaction if they post a reply to an individual outside their own group. We introduce a new measure of an individual's conformity to their own group based on contextualized embeddings of tweet texts. We call this the **Group Conformity Score (GCS)** of a user which measures how similar a user's tweets are to their own group as opposed to the tweets by users of the other group. Polarization is the sum of GCS over all the users, weighted by the inverse of their group size. We then unveil the heterogeneous effects of intergroup interactions on a person's group conformity over different pandemic related events using a meta-learning framework. We examine the heterogeneities in the effect with respect to religion, topics, emotions, engagement, and ego-network features. Finally, we decompose the differences between the treatment effects for the two religious groups into the treatment effects on change in topics of discussion and on change in emotions.

Existing studies measure linguistic polarization between groups on social media based on certain dimensions of tweet text such as stance [12] or contextualized embeddings of specific keywords [14]. Our approach is inspired by Gentzkow et al. [20] who propose a bag-of-words (BOW)-based metric to measure partisanship in congressional speech using the entire content. This metric overcomes finite sample bias resulting from phrases that a group might simply mention by chance. This metric has further been used to examine polarization on twitter in the context of mass shooting incidents in the US [13]. One limitation to the bag-of-words representation is that it does not take the context or the synonymy in two phrases into account. Our new contextualized-embeddings-based measure (GCS) can address this by capturing different dimensions of linguistic polarization more meaningfully.

Our work is motivated by the contact hypothesis which states that intergroup contact can reduce prejudice towards the outgroup when groups engage in equal status contact in the pursuit of common goals and in the presence of intergroup cooperation under a favorable institutional environment [2]. Thus, intergroup interaction can lead to a better understanding of outgroup perspectives and lead to cross-group friendships [10, 32]. At a broader level, this may facilitate national integration but may have the opposite effect in polarized settings [7]. In the political arena, Levendusky and Stecula [26] experimentally demonstrate that cross-party discussions decrease affective polarization (or dislike of outgroup individuals)

*Conference'17, July 2017, Washington, DC, USA*
2023. ACM ISBN 978-x-xxxx-xxxx-x/YY/MM...$15.00
https://doi.org/XXXXXXX.XXXXXXX

between Republicans and Democrats in the US. This decrease is conditional on conversation topics not involving disagreements [35]. The effects of intergroup contact might vary for majority and minority groups with possibly weaker effects for minorities [36]. Interestingly, intergroup contact focusing on commonalities between majority and minority groups can lead the minority to perceive the majority group as fairer than they are [34]. In the Indian context, Lowe [27] randomly assign individuals from different caste groups to the same (collaborative contact) or opposing (adversarial contact) cricket teams. They find that collaborative contact increases cross-caste friendships while adversarial contact has the opposite effect—thus highlighting the importance of the setting. In the online context, intergroup conversations between Hindus and Muslims on Whatsapp are found to decrease prejudice against Muslims [28]. We add to this evidence by going beyond prejudice and focus on group conformity in tweet text for both majority and minority groups, and examine the heterogeneous effects of intergroup contact.

In line with the above discussion, we hypothesize that in general intergroup interaction should decrease Group Conformity Score (GCS), and thus polarization. However, such interaction should be less likely to decrease GCS for individuals with already entrenched positions and who might be less receptive to outgroup perspectives. Further, when individuals in the minority group are disproportionately affected by an event, we expect intergroup interaction to amplify GCS for them. We expect the opposite effect for the unaffected majority group who might become sympathetic to minority issues due to interaction. Finally, for politically salient events, intergroup interaction should increase polarization for individuals having a high predisposition towards political discussions and who might have conflicting ideologies.

## 2 DATA

### 2.1 COVID-19 Tweets India

We use the "Global Reactions to COVID-19 on Twitter" data collected by Gupta et al. [22]. The core data comprise over 132 million english language tweets from more than 20 million unique users using 4 keywords—"corona", "wuhan", "nCov", and "COVID". The tweets were posted during January 28, 2020–January 1, 2021.[1] We use the India sample of the data, i.e. tweets pertaining to or originating from India.[2] Hydrator application is used to obtain complete information on tweets from their IDs (collected on May 4, 2021). Out of a total of 6,166,152 tweet IDs, full data for 5,459,402 tweets could be collected representing an attrition rate of 11.46%. This is due to deletion of some of the tweets and accounts by the collection date. The tweets in the data are posted by 871,203 unique users. The tweets are cleaned by removing mentions, hyperlinks and extra whitespaces. The data contains information on user name, their account creation date, number of friends and followers, whether a tweet is a retweet or a reply. It also contains information on

---

[1]The dataset of tweet IDs is publicly available at https://doi.org/10.3886/E120321V6.
[2]This restriction is imposed by mapping the user location attribute on twitter to country using GeoNames' cities15000 geographic database available at http://download.geonames.org/export/dump/cities15000.zip. The place field and the user location field are mapped to the India by matching them with a dictionary of cities and states in India. For places that could not be mapped to India using the previous step we use Nominatim—a search engine used for OpenStreetMap (OSM). This gives complete address of a place and allows us to remove tweets posted from outside India.

five psycho-linguistic attributes for each tweet indicating the intensity of valence, anger, fear, sadness, and joy extracted using CrystalFeel—"a collection of machine learning-based emotion analysis algorithms for analyzing the emotional-level content from natural language".[3]

### 2.2 Events

To select events for subsequent analysis, we scan the news reports for the important events that took place in 2020. We check which events became a topic of discussion within COVID-related discourse by scanning for event-related key-phrases in the tweets in our dataset. We present the details in Appendix A Table 3. We find that among all news headlines, the subsample in Table 1 were among the most popular COVID-related subjects on Twitter. And we focus our attention on these events in the rest of the paper.

## 3 METHODOLOGY

To study the treatment effect of Intergroup Interactions on the Group Conformity of an individual, we first augment our dataset to include all the variables of interest. We describe these in Sections 3.1–3.3 and then describe the steps for treatment effect estimation in Section 3.4.

### 3.1 Inferring Religion

Since the data on religious identity is not available on Twitter, we use usernames as a proxy of their religious identities. In India, names are highly correlated to the group identities and we leverage tcharacter sequence-based machine learning models from Chaturvedi and Chaturvedi [11] to obtain religion estimates.[4] The religion of each user is classified as Muslim (also referred to as minority as Muslims are the largest minority group in India comprising 14% of total population of India according to Census 2011) or non-Muslim (alternatively referred to as Hindus or majority group comprising 80% of total population).

We transliterate the names from indic languages Hindi, Bengali, Gujarati, Punjabi, Malayalam, Kannada, Tamil, Telugu, Oriya, Marathi, Assamese, Konkani, Bodo, Nepali, and Urdu to English using Indic-trans tool [8].[5] We drop verified users from this data to remove influential individuals/organizations from the data set. To further distinguish individuals from organizations and to weed out fake names, we construct a name part dictionary by using person names from a 3% random sample of eligible voters from Indian electoral rolls and Rural Economic & Demographic Survey (REDS) data collected by the National Council of Applied Economic Research. For every twitter user, only name parts that occur in the constructed name part dictionary are retained. We further manually scan the user names that either have more than 20,000 followers or those who tweeted more than 60 times or user names that contain any of the organization-related keywords provided in Appendix B Table 4 and drop the tweeters having non-personal names from this list.[6]

---

[3]See https://socialanalyticsplus.net/crystalfeel/.
[4]We use the Single Name SVM model to obtain the muslim-score as recommended.
[5]Available at https://github.com/libindic/indic-trans.
[6]To check for the possibility that our results might be influenced by bots, we use the recent lists TwiBot-20 [18] and TwiBot-22 [17]. We find that less than 0.5% of users in our data are listed as bots and contribute to 0.63% of the tweets.

**Table 1: Description of COVID-Related Events Highly Discussed in COVID Tweets India subset in year 2020**

| Event | Date | Description |
|---|---|---|
| Janata Curfew | Mar 22 | A day long curfew announced by the government, for all citizens barring essential services to curb the pandemic (and possibly to prepare for future lockdown). |
| Tablighi | Mar 31 | Tablighi Jamaat is a congregation of Muslims where attendees share food, sit close together, and preach religious teachings. Despite an ongoing ban on public gatherings, this assembly took place in Delhi and became a COVID-19 super-spreader event. Media reports about several participants (who were being quarantined post-gathering) spitting on attending doctors and healthcare personnel fueled Islamophobic sentiments across social media along with a viral hashtag #Corona Jihad. Much later, the Supreme Court slammed media outlets for communalizing the incident. |
| Migrant Deaths | May 8 | An estimated 10 million workers were forced to undertake long arduous journeys back home on foot after losing jobs due to abrupt pandemic-related lockdown and suspension of train services. 16 of them were killed by an empty goods train while they were sleeping on the tracks on this day. |
| Coronil Launch | Jun 22 | Indian multinational conglomerate Patanjali spearheaded by popular Indian yoga guru Ramdev launched an ayurvedic remedy claiming that it cures the COVID-19. This claim was not backed by any clinical data, yet the remedy was approved by Ministry of Ayush (for traditional medicine). The release received massive praise from some and harsh criticism from others on social media. Amidst controversy, the ministry ordered halt on the sale and advertising but later allowed it to be marketed as an immunity booster. However, it was completely banned by several state governments who considered it a fake medicine. |
| Exam Satyagraha | Aug 23 | All India Students' Association called for one-day hunger strike and satyagraha against the government's decision to conduct national-level examinations in-person. The reasons to oppose were personal health risks owing to the COVID-19 pandemic and logistics related challenges faced by students owing to the lockdowns and suspension of public transportation. More than 4000 students participated and multitudes showed support via social media. |
| GDP Contraction | Aug 31 | Indian government announced the biggest economic slump in GDP that India had seen in 24 years. |
| BJP Bihar Manifesto | Oct 22 | The ruling political party (BJP) promised free vaccines for everyone in Bihar in its Assembly election manifesto. People from both the religious groups over social media condemned this, reacting with the hashtag #VaccineForVotes. |

The model from [11] is trained on names obtained from official records, while the twitter names sometimes also include additional characters to reflect other personality attributes of the user. The name classification exercise also depends on the distribution of names in the specific domain the algorithm was trained on. Therefore, we expect a domain shift while applying the model on our data. To address the noisy classification issue, We experiment with two alternative decision boundary thresholds of the muslim-score returned by SVM model.[7] In the first case, a threshold of zero is used. This means that all users with muslim-score above 0 are classified as muslims and non-muslim otherwise. In the second, we split all the names into equal-width bins after sorting them on the basis of the muslim-score. We used 20 bins of width 0.1 each (The first bin encapsulates names with score below -0.9, and the last with 0.9 and above. We then randomly sample 50 names from each bin and manually annotate them. Thereafter, we plot roc-auc curves and analyze the gmean and Youden statistics to choose a threshold of 0.3 as the decision boundary. That is, if the muslim-score is greater than 0.3, we classify a name as muslim, and non-muslim otherwise.[8] We also find a very common non-muslim name Abhishek as being classified Muslim, we manually classify this as non-muslim.

## 3.2 Measuring Polarization

Using the estimates of religious identities of tweeters, we next measure the polarization in terms of conformity of each user to their religious group.

### 3.2.1 *Polarization via Bag-of-Words*. We first consider the leave-out estimator of phrase partisanship proposed by Gentzkow

et al. [20].[9] We compute daily user-level polarization [13] or the bag-of-words-based Group Conformity Score ($GCS_{i,d}^{BOW}$) as follows:

$$GCS_{i,d}^{BOW} = \hat{q}_{i,d} \cdot \hat{\rho}_{-i,d}$$

Where $\hat{q}_{i,d}$ is the vector of token (unigram and bigram) frequencies normalized by the sum of all token counts for user $i$ on day $d$ and $\hat{\rho}_{-i,d}$ is the vector denoting the sum of normalized token frequencies across all users in $i$'s group $g_i \in \{Muslim(M), non - Muslim(NM)\}$ while leaving $i$ out, relative to users in the other group $\tilde{g}_i$. For token $t$, $\hat{\rho}_{-i,d}^t = \sum_{j \in g_i - \{i\}} \hat{q}_j^t / (\sum_{g_i - \{i\}} \hat{q}_j^t + \sum_{\tilde{g}_i} \hat{q}_j^t)$.[10] Intuitively, this captures similarity in phrase usage for user $i$ with their group members relative to the similarity with the other group. The daily polarization is then estimated as the following average:

$$\hat{\pi}_d^{LO,BOW} = \frac{1}{2} \sum_{g \in \{M,NM\}} \frac{1}{|g|} \sum_{i \in g} GCS_{i,d}^{BOW}$$

### 3.2.2 *Polarization via Contextualized Embeddings*. The bag-of-words-based polarization estimator ignores the larger context of a tweet and can have several limitations. Firstly, distinct words (for example, greetings such as *salaam* vs. *namaste*) used by two groups conveying the same underlying message will contribute positively towards the polarization estimate. Secondly, if two users have different stance on a given issue while having broadly similar phrase usage (for example, *Coronil cures Covid* vs. *Coronil does not cure Covid*), the BOW estimator will consider them to be similar.

We address these by computing the contextualized-GCS score or simply $GCS_{i,d}$ for user $i$ on day $d$. For this, we first map all

---

[7]Both the thresholds lead to qualitatively similar end results.

[8]See Appendix C, Figure 5.

[9]Before applying this estimator, we lower-case the tweets, remove stopwords (see Appendix D for the stopwords list) and punctuations, and stem words using the NLTK's Snowball Stemmer. Removing stopwords leads to dropping 1,007 user-day observations comprised entirely of stopwords and punctuations.

[10]We only consider tokens $t$ which are used by at least two users.

the tweets to a 768-dimensional vector space using a sentence-transformer—specifically, the pretrained *all-mpnet-base-v2* model [33]. This model is fine-tuned on over a billion sentence pairs from diverse domains and has shown state-of-the-art results on semantic search and sentence embedding tasks.[11] We then average these embeddings at the user-day level $u_{i,d}$. Next, we compute daily centroids for both the groups by taking the mean of $u_{i,d}$ across all users in a group. Analogous to the leave-out estimator, we first adjust the group centroid by subtracting $u_{i,d}$ from the group centroid before computing $GCS_{i,d}$. Thereafter, we compute the distances between $u_{i,d}$ and both the centroids. Finally, $GCS_{i,d}$ is computed as the Euclidean distance from the other group's centroid relative to their own adjusted centroid. We use the following formula:

$$GCS_{i,d} = \frac{\|u_{i,d} - \frac{1}{|\tilde{g}_i|} \sum_{j \in \tilde{g}_i} u_{j,d}\|}{\|u_{i,d} - \frac{1}{|g_i - \{i\}|} \sum_{j \in g_i - \{i\}} u_{j,d}\| + \|u_{i,d} - \frac{1}{|\tilde{g}_i|} \sum_{j \in \tilde{g}_i} u_{j,d}\|}$$

Higher values of $GCS_{i,d}$ correspond to greater conformity of a user to their own group. The daily polarization $\hat{\pi}_d^{LO}$ is computed by aggregating this measure across users in the two groups as before:

$$\hat{\pi}_d^{LO} = \frac{1}{2} \sum_{g \in \{M,NM\}} \frac{1}{|g|} \sum_{i \in g} GCS_{i,d}$$

## 3.3 Discussion Topics During COVID-19

To gain a deeper understanding of major topics discussed around Covid-19 events and to include them as covariates for examining the effect of intergroup interaction on change in group conformity, we perform topic modeling over the tweets. We use the subset of tweets considered for treatment effect estimation across all the events. We leverage contextualized embeddings for this task as well. We first drop the duplicated tweets so that retweeting does not affect topic assignment. We then cluster the tweets' contextual embeddings obtained using sentence transformer model *all-mpnet-base-v2* using the k-means clustering algorithm [21, 37].[12] For inferring representative topic labels, we preprocess the tweets by first lower-casing them. Since the entire dataset comprises COVID-specific tweets, we remove COVID and its synonyms for a more meaningful inference of topic labels. We also replace different vaccine names with the word vaccine, remove mentions, urls, numbers, special html entities such as &amp and &quot, punctuation, and extra spaces. We then transform each tweet by joining together commonly occurring multi-word phrases in that tweet using the Gensim phrase model [29]. We then concatenate all the tweets within a topic as a single document. Finally, we compute class-based TF-IDF defined as:

$$cTF\text{-}IDF_i = \frac{t_i}{w_i} \cdot log \frac{m}{\sum_{j=1}^{n} t_j}$$

Where, $t_i$ is frequency of a word/phrase within the $i^{th}$ topic and $w_i$ the total number of phrases in the topic. The total number of tweets (or unjoined documents) is $m$, and is normalized by the number of occurrences of the word/phrase across all $n$ topic clusters.

We identify the following topics: General COVID response, COVID prevention, COVID News/statistics (general), COVID news/statistics (state-specific), Socio-Economic, Political-Religious, and China & Global. We do a qualitative and quantitative analysis of the topic clusters and find that the COVID-specific topics are very similar and merge them into a single topic COVID Response to get final four topics.[13] We provide 50 most representative phrases associated with each of these topics and and a sample of tweets associated with each topic in Appendix F Tables 5 and 6 respectively.

## 3.4 Conditional Average Treatment Effect

In this section we describe our methodology to answer the question *do intergroup interactions change a user's conformity to their group?*. We first describe the treatment and outcome variables:

### 3.4.1 *Treatment: Intergroup Interaction*. For each event, we look at all tweets before the event. We consider tweets that are replies, and check the religion of the user posting the reply and that of the user being replied to. Our treatment variable *interact* is a binary indicator that equals 1 if a user replies to someone outside their group at least once, and 0 otherwise.[14]

### 3.4.2 *Outcome*.

*Change in GCS.* For each event, we define an event window of n days before and after it. We compute the n-day mean of $GCS_i$ for each user $i$ over the pre-event and post-event windows.[15] Finally, we take $\Delta GCS_i = \overline{GCS_{i,post}} - \overline{GCS_{i,pre}}$ as the difference in the averages post and pre-event. We choose the window size to be large enough to balance the daily fluctuations and small enough to rule out other events influencing the outcome.

*Change in Topics and Emotions.* We also estimate the treatment effect of intergroup interaction on change in topics and emotions after each event. This helps in decomposing the differential effects of intergroup interaction on $\Delta GCS_i$ in terms of differential effects on changes in topics and emotions across religions. We again take the mean difference in these variable during post and pre-event windows for each user.

### 3.4.3 *Pre-treatment Covariates*. We consider 30 days pre-event window and compute the following covariates for adjustment: 30-day averages of $GCS$, emotion intensities for valence, anger, fear, sadness, and joy; ego-network features such as friends and followers counts; engagement features such as tweet frequency, average number of times a user's tweets were retweeted and the fraction of replies among tweets; the number of days lapsed since account creation to event date; muslim-score as given by the religion classifier; and lastly the fraction of user tweets in the pre-treatment period assigned to each topic. We use inverse hyperbolic sine transformation (arcsinh) for friends and followers-counts, tweet frequency and average retweets, as these are right skewed. Arcsinh approximates logarithmic transformation while allowing us to retain zero-values. In We then normalize all the covariates and the outcome variable.

---

[11]For more information, see https://huggingface.co/sentence-transformers/all-mpnet-base-v2.

[12]We choose 7 as the optimal number of clusters based on manual scanning and the elbow method heuristic by plotting inertia against the number of topics over 3 to 10 topic clusters.

[13]The qualitative analysis is based on most frequent words associated with each topic while quantitative analysis is based on cluster mean and mean of pairwise cosine distances across clusters. Our results for metalearners in Section 4.2 remain the same with and without merging topic clusters.

[14]The users who never reply to anyone are dropped from further analysis

[15]Users who do not tweet during any window are dropped.

The descriptive statistics for the final event-level dataset are provided in Appendix H Table 9.

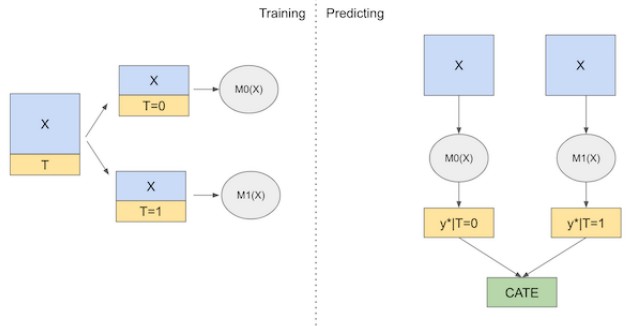

Training ‖ Predicting

**Figure 1: T-Learner Framework. Source-Alves [3]**

*3.4.4* **T-Learner**. Given the outcome $Y$ ($\Delta GCS$) and specific values of pre-treatment covariates $X = x$, Conditional Average Treatment Effect (CATE) is defined as:

$$\tau(x) = \mathbb{E}[Y(1) - Y(0) \mid X = x] = M_1(X) - M_0(X)$$

Where, $Y(1)$ is the $\Delta GCS$ observed for an individual in the Treatment group ($T = 1$), i.e. if they interact outside their group and $Y(0)$ is $\Delta GCS$ for the same individual considering they belong to the control group ($T = 0$). Here, we can only observe either $Y(1)$ or $Y(0)$ for any given individual. Since our treatment variable is discrete, we can leverage meta-learning algorithms—which help us estimate the response functions $M_1(X)$ and $M_0(X)$, and thus CATE. In particular, we use T-Learner [25] which consists of the following two stages as depicted in Figure 1:

(1) **Training Stage**: In the first stage, we learn approximations of response functions $\hat{M}_1$ and $\hat{M}_0$ using observations from the treatment and control groups respectively.
(2) **Prediction Stage**: In the second stage, we estimate Individual Treatment Effect ITE for $i^{th}$ user using predictions from $\hat{M}_1$ and $\hat{M}_0$ over the complete set of observations in the test set as:

$$\hat{\tau}(x_i) = \hat{M}_1(x_i) - \hat{M}_0(x_i)$$

*Implementation Details.* For measuring the outcome, we use a window size of 7 days post and pre-event. We use nested Lasso regression with 10-fold cross validation (CV). We first split the event-specific subsets for each treatment group into 10 folds. For each iteration, we further split the training fold and run Lasso with L2 regularization, using 10-fold CV. We report the average MSE and R-squared along with the average treatment effect.[16]

---

[16]Base learners such as support vector regression (SVR), random forest, Ridge, or RANSAC regression with grid-search for hyperparameter tuning lead to lower performance metrics. We also experiment with window sizes of 5 or 10 days and find broadly similar trends. We get qualitatively similar results using X-learner [25].

## 4 RESULTS

Here, we first discuss a qualitative analysis based on our proposed metric *GCS* in Section 4.1 and then the results from treatment effect estimation in Section 4.2. Finally, we explain the differences in the treatment effect across the two religious groups in terms of the effect on changes in topics and emotions using a decomposition approach in Section 4.3.

## 4.1 Qualitative Content Analysis based on GCS

To compare the BOW and contextualized-embeddings-based estimator, Figure 2 plots the seven-day exponential moving average of polarization trends using $\hat{\pi}^{LO,BOW}$ and $\hat{\pi}^{LO}$. We find similar trends using both the measures with the Pearson's correlation of 66.87%. However, the fluctuations in BOW polarization are more pronounced. The polarization increases during the Tablighi incidence on March 31, 2020, which was marked by increased Islamophobic sentiments. The highest peaks are during the Muslim festivals—the beginning of holy month of Ramadan and its culmination in Eid-ul-fitr. There are also smaller peaks during the Muslim festivals of Eid-ul-zuha and Eid-e-Milad. We find that the Muslim tweets during these festivals are mostly greetings and well wishes while non-Muslim tweets discuss a variety of subjects. We do not find any peaks around non-Muslim festivals.

To understand whether *GCS* provides a more meaningful measure compared to $GCS^{BOW}$ we examine tweets from users in both religious groups having high and low values of these measures up to seven days post each event.[17] As expected, the tweets from users in one group having high group conformity are similar to tweets by users in the other group having low group conformity. This holds even without exactly matching but semantically similar tweets in case of *GCS*. The highest $GCS^{BOW}$ tweets for the majority group often comprise a few common words. This is consistent with the results in Appendix E, Figure 6 in which we examine the relation of tweet length with $GCS^{BOW}$ and *GCS*. We find that the average tweet length is low at extreme values of $GCS^{BOW}$ while this relation is weak in case of *GCS*.

To qualitatively examine each group's tweeting patterns during each event, we focus on high *GCS* user tweets. Starting with the pre-lockdown **Janata Curfew**, we notice hostile attitudes against China and appreciation for frontline workers among non-Muslims, while Muslims predominantly share news related to Kashmir (a Muslim-majority state) and Muslim-majority countries. In context of the **Tablighi** incidence, non-Muslims promoted the Indian prime minister's plea to light candles in a show of unity in fight against COVID-19 at 9 p.m. for 9 minutes, whereas Muslim tweets express anger over Islamophobia spread by Indian media in the context of this event. A noteworthy example, in the aftermath of the Tablighi incident, is of a non-Muslim user supporting Muslims: *"The manner in which media showed propaganda against tabliqi jamat & corona jihad etc but didn't shown Bombay HC judgements which said tablighi's were made scapegoats, same way they will show propaganda against @Tweet2Rhea & @deepikapadukone but will not show u the judgments later"*. *GCS* (0.497 or $9^{th}$ percentile) correctly identifies

---

[17]Appendix G, Tables 7 and 8 report samples of 5 tweets from high *GCS* and $GCS^{BOW}$ users in each group.

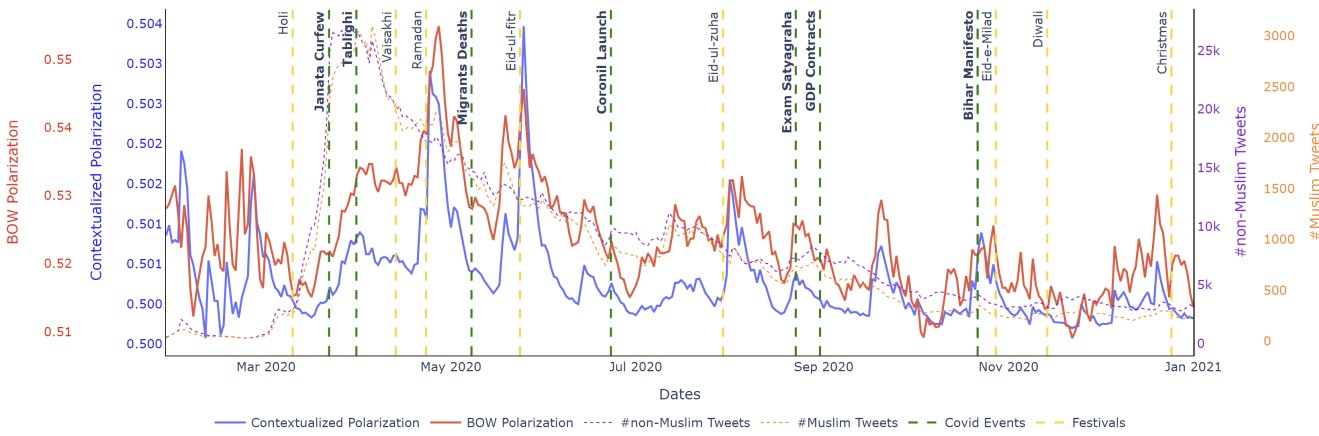

**Figure 2: 7-day Exponential Moving Average of daily polarization estimated using contextualized approach $\hat{\pi}^{LO}$ vs. bag-of-words approach $\hat{\pi}^{LO,BOW}$ along with the number of COVID-related tweets by both religious groups. The COVID-related events are marked with green vertical lines and major festivals are marked with yellow vertical lines.**

low group conformity for this user while $GCS^{BOW}$ (0.84 or $99^{th}$ percentile) fails to do so.

After **Migrant Deaths** there's little discussion on the plight of migrants among high $GCS$ non-Muslims. When a discussion occurs, it captures two viewpoints—one is the suffering of migrants and the second is the increasing risk of COVID spread and economic issues resulting from migration. On the other hand, high $GCS$ Muslims express anger against the government and media for not covering the issue. After **Coronil Launch**, non-Muslims express relief against COVID and pride in the Indian Ayurvedic medicine, though a few express skepticism as well. On the other hand, several high $GCS$ Muslims label it a fake drug. In addition, Muslim discourse remains centered around Islamophobia and the bigotry of news media in coverage of Tablighi vs. government approval of Rath Yatra—a Hindu religious congregation.

Post the call for **Exam Satyagraha**, high $GCS$ non-Muslim discourse is more varied with some concerns regarding increased COVID risk due to in-person examination while high $GCS$ Muslims harshly criticize the decision of in-person exams. This is perhaps due to Muslims being the poorest religious group in India and might find it logistically harder to attend in-person exams. Similarly, during **GDP Contraction** high $GCS$ non-Muslim discourse remained more varied while Muslim tweets express criticism towards the government over GDP decline. Finally, after the release of **Bihar Manifesto** by the ruling party BJP, high $GCS$ non-Muslim discourse focuses on general COVID-related news with some criticizing the *vaccine for vote* clause in the manifesto. This criticism appears to be unanimous among the high $GCS$ Muslim tweeters.

## 4.2 Effect of Interaction on Change in GCS

In this section, we examine the results from T-learner. Table 2 shows the average effect $\hat{\tau}_{All}$ of intergroup interaction on change in overall $GCS$ and also separately for Muslims $\hat{\tau}_{Muslim}$ and non-Muslims $\hat{\tau}_{non-Muslim}$ across all the events. Appendix I, Figure 7 shows corresponding distributions of individual treatment effects for Muslims

and non-Muslims. We find that intergroup interaction decreases overall $GCS$ (or $\hat{\tau}_{All} < 0$) for all events except GDP Contraction for which there is an increase in $GCS$ ($\hat{\tau}_{All}$ = 0.05 standard deviations or s.d.). In other words, talking to people from the other group generally contributes to a decrease in polarization. The strongest negative effect among these is for the Tablighi incident (-0.16 s.d.). In contrast, while intergroup interaction decreases average $GCS$ for Muslims after all the other events, the effect is positive for the Tablighi event (0.04 s.d.) which was a highly communal event followed by increasing islamophobia in India. This suggests that intergroup interaction amplifies the polarizing effect of such events for the affected minorities. Notably, the negative effect for Muslims after GDP Contraction is not statistically significant while all the other coefficients we discuss are statistically significant at the 1% level of significance. The strongest negative effect of intergroup interaction on $GCS$ for Muslims is in case of Janata Curfew (-0.24 s.d.) and after the release of Bihar Manifesto (-0.18 s.d.).

We examine the heterogeneity in the treatment effect (TE) by regressing the treatment effect $\hat{\tau}$ on standardized pre-treatment covariates for each event. The complete results are reported in Appendix I, Figure 8 but we highlight the most important findings here. We find a high positive correlation between pre-treatment $GCS$ and $\hat{\tau}$. In other words, the decline in $GCS$ due to intergroup interaction is stronger for people with an already low $GCS$. This is especially true in case of Bihar Manifesto which is a highly political event, and on which Hindus and Muslims might have divergent perspectives. However, this positive correlation breaks down in case of the Tablighi event. Among the topics, $\hat{\tau}$ is more negative after launch of Coronil remedy for people who were initially more engaged in *COVID response* discussion, and hence, might have shared concerns related to this. The other notable topic is Politics-Religion with which $\hat{\tau}$ has a highly positive correlation in case of Exam Satyagraha and Bihar Manifesto both of which are political events. Specifically, Exam Satyagraha was called for by the left wing student organization AISA when the right wing ruling government

**Table 2: Effect of intergroup interaction on $GCS$ estimated using Meta-learner using Lasso with 10-fold CV. We separately report average treatment effects for each group and report bootstrapped standard errors in parentheses.**

| Event | #Users | $\hat{M}_0$ | | $\hat{M}_1$ | | $mean(\Delta GCS)$ | | Treatment Effect | | | |
|---|---|---|---|---|---|---|---|---|---|---|---|
| | | $R^2$ | MSE | $R^2$ | MSE | Control | Treated | $\hat{\tau}_{All}$ | $\hat{\tau}_{Muslim}$ | $\hat{\tau}_{non-Muslim}$ | $\Delta\hat{\tau} = \hat{\tau}_{Muslim} - \hat{\tau}_{non-Muslim}$ |
| Janata Curfew | 4671 | 0.091 | 0.892 | 0.013 | 1.088 | 0.003 | -0.027 | -0.055 | -0.240 | -0.038 | -0.203 |
| | | | | | | | | (0.003) | (0.009) | (0.003) | (0.010) |
| Tablighi | 6946 | 0.344 | 0.642 | 0.345 | 0.719 | -0.001 | 0.006 | -0.160 | 0.044 | -0.181 | 0.225 |
| | | | | | | | | (0.002) | (0.006) | (0.002) | (0.007) |
| Migrant Deaths | 6387 | 0.116 | 0.856 | 0.068 | 1.080 | 0.002 | -0.013 | -0.093 | -0.097 | -0.093 | -0.004 |
| | | | | | | | | (0.002) | (0.008) | (0.002) | (.008) |
| Coronil Launch | 4622 | 0.240 | 0.753 | 0.099 | 0.930 | 0.007 | -0.035 | -0.050 | -0.082 | -0.047 | -0.035 |
| | | | | | | | | (0.002) | (0.009) | (0.003) | (0.010) |
| Exam Satyagraha | 3497 | 0.185 | 0.782 | 0.118 | 0.994 | 0.012 | -0.053 | -0.053 | -0.027 | -0.055 | 0.028 |
| | | | | | | | | (0.002) | (0.013) | (0.002) | (0.013) |
| GDP Contraction | 3792 | 0.146 | 0.853 | 0.042 | 0.931 | -0.012 | 0.051 | 0.050 | -0.010 | 0.056 | -0.066 |
| | | | | | | | | (0.004) | (0.012) | (0.004) | (0.010) |
| Bihar Manifesto | 1989 | 0.066 | 0.930 | -0.025 | 0.984 | 0.007 | -0.028 | -0.087 | -0.181 | -0.080 | -0.101 |
| | | | | | | | | (0.005) | (0.033) | (0.006) | (0.032) |

announced the decision to conduct exams. Among the emotions, we find a high positive correlation of $\hat{\tau}$ with Anger in case of the communally charged Tablighi event indicating that the intergroup interaction increased $GCS$ for people who expressed more anger earlier.

## 4.3 Decomposition Analysis

Given that the effects of interaction on change in $GCS$ exhibit substantial heterogeneity across the two religious groups, a natural question is—*what is the contribution of topics and emotions towards explaining these differences?* Importantly, emotions and topics are also computed as properties of the tweet text, and changes in $GCS$ partially embody changes in these attributes.[18] Therefore, we decompose the mean of $\Delta\hat{\tau} = \hat{\tau}_{Muslim} - \hat{\tau}_{non-Muslim}$ into the effect on each topic and emotion. We use the Oaxaca-Blinder method [9, 30] to decompose the differences at the *mean* into *explained* and *unexplained* components and further into contributions of individual covariates to explained differences.

Given $\bar{\bar{\tau}}_g = \sum_x \beta_g^x \bar{\bar{\tau}}_g^x$, $g \in \{Muslim(M), non - Muslim(NM)\}$, where $\beta_g^x$ are the regression coefficients and $\bar{\bar{\tau}}_g^x$ are mean values of covariates (the average treatment effects on $x \in topics \lor emotions$):

$$\Delta\bar{\bar{\tau}} = \bar{\bar{\tau}}_M - \bar{\bar{\tau}}_{NM}$$
$$= \sum_{x \in topics \lor emotions} \beta_M^x \bar{\bar{\tau}}_M^x - \beta_{NM}^x \bar{\bar{\tau}}_{NM}^x$$
$$= \sum_{x \in topics \lor emotions} \underbrace{\beta_{NM}^x (\bar{\bar{\tau}}_M^x - \bar{\bar{\tau}}_{NM}^x)}_{\text{Explained}} + \underbrace{\bar{\bar{\tau}}_M^x (\beta_M^x - \beta_{NM}^x)}_{\text{Unexplained}}$$

The explained component captures what part of the mean difference in treatment effect across Muslims and non-Muslims is due to a differential shift in topics and emotions due to intergroup interaction. The residual or unexplained component captures to what extent the marginal effect of each covariate on the outcome is different across the two groups, given that they have the same explanatory attributes.

Figure 3 shows the aggregate decomposition into the explained and unexplained components. We observe the largest negative $\Delta\bar{\bar{\tau}}$

---

[18]Appendix Figures 9–17 show the distribution of treatment effects on these explanatory variables across Muslims and non-Muslims for each event.

in case of Janata Curfew (-0.2 s.d.), and Bihar Manifesto (-0.1 s.d.) and the explained component of these differences are estimated at 46.5% and 64.6% respectively. We also find a highly positive difference (0.2 s.d.) in case of Tablighi incident and the explained component of this difference is 7.8%. The difference in case of GDP Contraction is (-0.07 s.d.). However, the explained component of this difference is not significantly different from 0. We also observe small negative $\Delta\bar{\bar{\tau}}$ for Coronil Launch (-0.04 s.d.) and Exam Satyagraha (-0.03 s.d.). For Coronil Launch, the covariates overexplain the difference with the explained component at 144%, while for Exam Satyagraha the explained difference is 94.9%.

Figure 4 further decomposes the explained component into the contributions of emotions and topics. We find that, for Janata Curfew, 81% of $\Delta\bar{\bar{\tau}}$ is explained by valence. This is because, for this event, $\beta_{NM}^{valence}$ is negative—i.e. an increase in valence due to intergroup interaction is associated with a decrease in $GCS$—and the difference $\bar{\bar{\tau}}_M^{valence} - \bar{\bar{\tau}}_{NM}^{valence}$ is positive. Additionally, 30% of $\Delta\bar{\bar{\tau}}$ is explained by the topic China & Global for this event. In contrast joy, sadness, and Politics-Religion topic have countervailing effects, i.e. they pull $\Delta\bar{\bar{\tau}}$ towards zero. In case of Tablighi incident, valence (9%) and joy (14%) explain an important share of $\Delta\bar{\bar{\tau}}$ while anger has a countervaling contribution (-11%). For both the politically salient events—Exam Satyagraha and Bihar Manifesto—the differential effect of intergroup interaction on Politics-Religion and Socio-Economic topics explain $\Delta\bar{\bar{\tau}}$.

## 5 CONCLUSION

Our study explores the complex relationship between intergroup interactions and polarization between religious groups on social media in light of events during the COVID-19 pandemic in India. We investigate whether these interactions serve as bridges that mitigate polarization or barriers that exacerbate it. We use a novel measure of group conformity based on contextualized embeddings to uncover a compelling narrative. Consistent with our hypotheses, intergroup interactions reduce polarization in general though this effect is less pronounced for individuals with strong group conformity (high GCS). Further, intergroup interactions increase the group conformity for the minority Muslim group individuals during the communal Tablighi event. Finally, in the context of political events such as Exam Satyagraha and Bihar Manifesto, intergroup

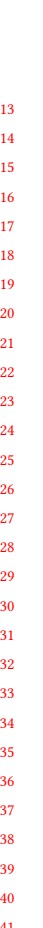

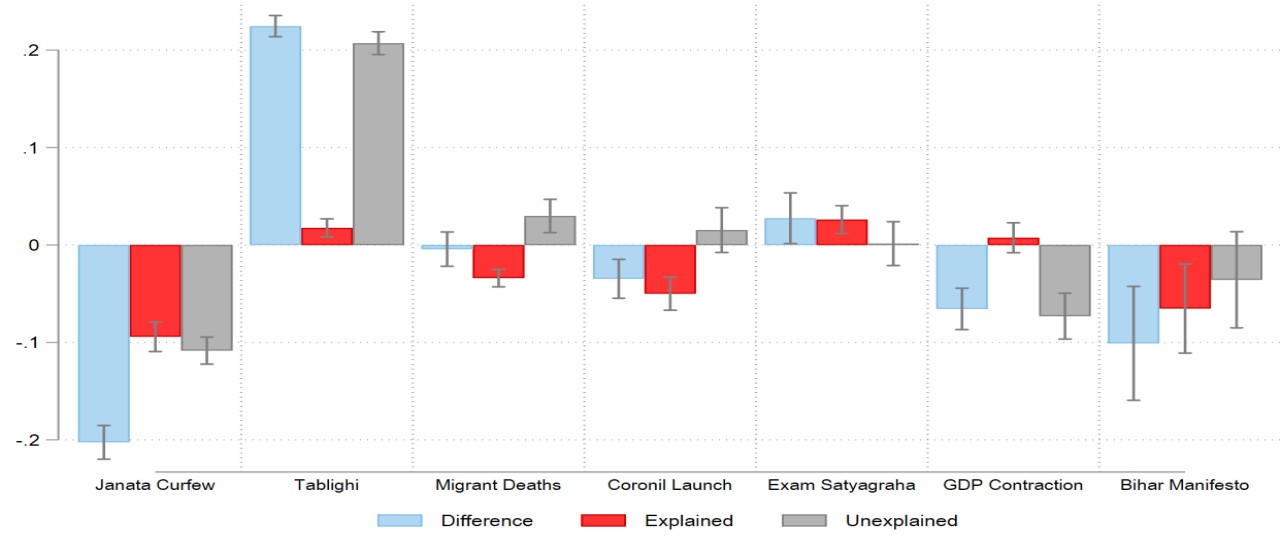

**Figure 3: Decomposition of difference in the effect of interaction on $GCS$ between Muslims and non-Muslims using Oaxaca-Blinder decomposition. The red bars show the extent to which the effect is explained by topics and emotions. The error bars represent 95% confidence intervals.**

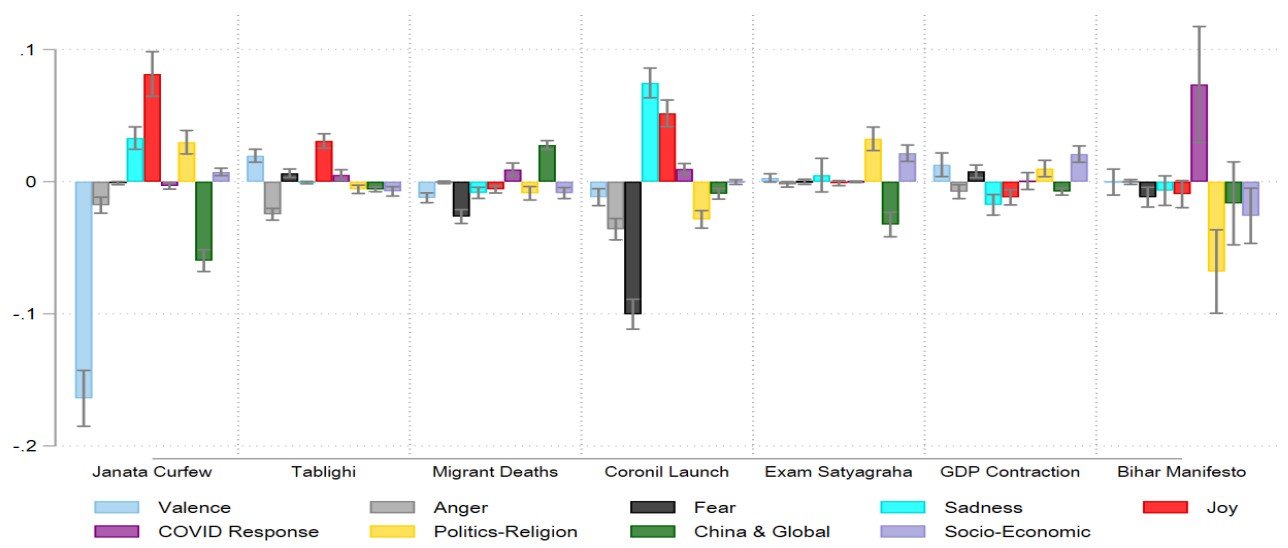

**Figure 4: Contribution of emotions and topics towards explained component of difference in effect of interaction on change in $GCS$ across Muslims and non-Muslims using Oaxaca-Blinder decomposition. Errors bars show 95% confidence intervals.**

interactions amplify polarization of politically inclined individuals. Additionally, we leverage a well-known decomposition method to explain the differences in average treatment effects of interaction on group conformity across the two religious groups in terms of effects on emotions and topics of discussion.

Our work highlights the importance of context-aware metrics and nuanced approaches in studying polarization dynamics. In line with previous works that utilize tweet content for predicting group identity [31], GCS score can also help improve existing name classification algorithms. Future studies could explore interventions and strategies to foster constructive intergroup interactions to encourage cross-group tolerance and peaceful coexistence in the face of societal challenges.

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

# Appendix

## A    EVENTS THAT MADE NEWS HEADLINES IN INDIA IN 2020

**Table 3: Events that made the news in 2020 in India and the frequency of tweets containing event-related key-phrases in 7 days post event date (inclusive).**

| EVENT | Date | Search String | Tweet Frequency | | |
|---|---|---|---|---|---|
| | | | All | non-Muslim | Muslim |
| Maoist attack | Feb 23 2020 | (?i)(maoist)\|(sukma)\|(chhattisgarh)\|(chattisgarh)\|(chateesgarh)\|(martyr) | 2 | 1 | 1 |
| Delhi Riots | Feb 23 2020 | (?i)(riot)\|(communal)\|(\bcaa\b)\|(nrc)\|(npr)\|(shaheen)\|(jaffraba)\|(jafraba)\|(delhipolice)\|(delhi police)\|(?=.*protest)(?=.*delhi) | 192 | 173 | 19 |
| Janata Curfew | March 22 2020 | (?i)(janata)\|(curfew)\|(janta)\|(junta) | 15032 | 13836 | 1196 |
| Tablighi | Mar 31 2020 | (?i)(tabliqi)\|(tablighi)\|(jamat)\|(jamaat)\|(coronajihad) | 10181 | 9020 | 1161 |
| Palghar Mob Lynching | Apr 16 2020 | (?i)(palghar)\|( mob )\|(#mob )\|(lynching)\|(lynched) | 1172 | 993 | 179 |
| Ramadan | Apr 23 2020 | (?i)(ramdan)\|(ramzan)\|(ramadan)\|(mubarak) | 2240 | 1117 | 1123 |
| Vizhag Gas Leak | May 07 2020 | (?i)((?=.*vizhag)\|(?=.*vizag)\|(?=.*vishakhap)\|(?=.*visakhapa)\|(?=.*gas))(?=.*leak)\|(?=.*leak)((?=.*lgpolymer)\|(?=.*lg polymer)) | 720 | 647 | 73 |
| Migrant Deaths | May 08 2020 | (?i)(\bmigrant) | 2838 | 2507 | 331 |
| Cyclone Amphan | May 15 2020 | (?i)(amphan)\|(cyclone) | 1618 | 1476 | 142 |
| idulfitr | May 24 2020 | (?i)(ul-fitr)\|(al-fitr)\|(\beid\b)\|(#eid)\|(idul)\|(fitr)\|(mubarak) | 1446 | 739 | 707 |
| Baghjan GasLeak | May 27 2020 | (?i)(baghjan)\|(leak)\|(oil( ?)india) | 117 | 108 | 9 |
| Vijaywada COVID facility fire | | (?i)(vijaywada)\|(vijayawada)\|(swarnahotel)\|(swarna hotel)\|((?=.*care facility)(?=.*fire)) | 291 | 261 | 30 |
| Cyclone Nisarg | Jun 2 2020 | (?i)(cyclone)\|(nisarg) | 2554 | 2353 | 201 |
| Agro-agrordinances | Jun 5 2020 | (?i)((?=.*ordinance)\|(?=.*act)\|(?=.*bill))((?=.*farm)\|(?=.*agro)\|(?=.*agricultur)(?=.*krishi)(?=.*kis[a]+n)) | 62 | 52 | 10 |
| Rail Suspension | Jun 25 2020 | (?i)(railway)\|(railband) | 46 | 44 | 2 |
| Sushant Singh Case | Jun 14 2020 | (?i)(sushant singh)\|(sushantsingh)\|(sushantsinghrajput) | 391 | 357 | 34 |
| China Border Skirmish | Jun 15 2020 | (?i)((?=.*border)\|(?=.*skirmish)\|(?=.*melee))(?=.*china)\|((gal)[vw](an))\|((martyr)\|(soldier)\|(troops))\|(gogra)\|(chinese aggression)\|(boycott( ?)china) | 2563 | 2349 | 214 |
| India UNSC seat | Jun 17 2020 | (?i)(?=.*india)(?=.*unsc)\|(indiainunsc)\|(unsc)\|(?=.*member)(?=.*un )\|(?=.*un)(?=.*security council) | 80 | 73 | 7 |
| Coronil Launch | Jun 23 2020 | (ayurved)\|(ayush)\|(ramdev)\|(coronil)\|(patanjali) | 6838 | 6366 | 472 |
| Rail Suspension | Jun 25 2020 | v(railway)\|(railband) | 410 | 392 | 18 |
| Ban Chinese Apps tiktok | Jun 29 2020 | (?i)(?=.*chinese)(?=.*ban)\|(chinese app)\|(chinese aap)\|(tiktok)\|(tik-tok)\|(digitalairstrike) | 599 | 548 | 51 |
| Vikas Dubey Kills | Jul 3 2020 | (?i)(dubey) | 150 | 136 | 14 |
| Pulwama | Jul 5 2020 | (?i)(pulwama) | 13 | 6 | 7 |
| Kargil Earthquake | Jul 5 2020 | (?i)(kargil)\|(earthquake) | 44 | 37 | 7 |
| Assam Flood | Jul 21 2020 | (?i)(flood)\|(brahmaputra)\|(assam) | 265 | 221 | 44 |
| NEP | Jul 29 2020 | (?i)(education policy)\|(nep2020)\|(\bnep\b) | 61 | 57 | 4 |
| Bakr-Eid | July 30 2020 | (?i)((?=.*ul)\|(?=.*al))((?=.*zuha)\|(?=.*adha))\|(bakr)\|(\beid\b)\|(#eid)\|(mubarak)((?=.*slaughter)\|(?=.*kill)\|(?=.*cruel)\|(?=.*sacrific))((?=.*animal)\|(?=.*goat)\|(?=.*lamb)) | 986 | 704 | 282 |

**Table 3: Events that made the news in 2020 in India and the frequency of tweets containing event-related key-phrases in 7 days post event date (inclusive).**

| EVENT | Date | Search String | Tweet Frequency | | |
|---|---|---|---|---|---|
| | | | All | non-Muslim | Muslim |
| Ram Temple Foundation | Aug 5 2020 | (?i)((ram )(temple\|mandir))\|(ram(temple\|mandir))\|(ayodhya)\|(babri)\|(jai(shri\|sri\|shree)ram)\|(jai (shri\|shree\|sri)ram)\|(babar)\|(?=.*demolish)(?=.*masjid) | 1043 | 872 | 171 |
| Air India crash | Aug 8 2020 | (?i)(airindia)\|(crash) | 643 | 558 | 85 |
| Vijaywada Fire | Aug 9 2020 | (?i)(vijaywada)\|(vijayawada)\|(swarnahotel)\|(swarna hotel)\|((?=.*care facility)(?=.*fire)) | 291 | 261 | 30 |
| Farmer Protest | Aug 9 2020 | (?i)((?=.*ordinance)\|(?=.*act)\|(?=.*bill)\|(?=.*protest))((?=.*farm)\|(?=.*agro)\|(?=.*agricultur))\|(krishi)\|(kis[a]+n) | 301 | 293 | 8 |
| BLRiot | Aug 11 2020 | (prophet)\|(?=.*riot)(?=.*b(e\|a)ng[a]?l((ore)\|(uru))) | 120 | 107 | 13 |
| Exam Satyagraha | Aug 23 2020 | (?i)(exam)\|(student) | 19384 | 17488 | 1896 |
| GDP Contraction | Aug 31 2020 | (gdp)\|(economy)\|(unemployment) | 6293 | 5561 | 732 |
| Chinese Apps ban pubg | Sep 3 2020 | (?i)(?=.*chinese)(?=.*ban)\|(?=.*app)(?=.*ban)\|(pubg)\|(chinese app)\|(chinese aap)\|(digitalairstrike) | 515 | 443 | 72 |
| Farm Bill Lower House | Sep 14 2020 | (?i)((?=.*ordinance)\|(?=.*act)\|(?=.*bill)\|(?=.*protest))((?=.*farm)\|(?=.*agro)\|(?=.*agricultur))\|(krishi)\|(kis[a]+n) | 442 | 380 | 62 |
| PM Modi's Birthday/#NationalUnemploymentDay | Sep 17 2020 | (?i)((?=.*unemployment)\|(?=.*b.?day)\|(?=.*birthday))(?=.*modi) | 268 | 203 | 65 |
| IPL | Sep 19 2020 | (?i)(cricket)\|(ipl) | 797 | 708 | 89 |
| Farm Bill Upper House | Sep 20 2020 | (?i)((?=.*ordinance)\|(?=.*act)\|(?=.*bill)\|(?=.*protest))((?=.*farm)\|(?=.*agro)\|(?=.*agricultur))\|(krishi)\|(kis[a]+n) | 634 | 530 | 104 |
| Bharat Bandh | Sep 25 2020 | (?i)(bharat band)\|(bharatband)\|((?=.*ordinance)\|(?=.*act)\|(?=.*bill)\|(?=.*protest))((?=.*farm)\|(?=.*agro)\|(?=.*agricultur))\|(krishi)\|(kis[a]+n) | 413 | 351 | 62 |
| Hathras victim dies | Sep 29 2020 | substring:(?i)(manishavalmiki)\|(manisha valmiki)\|(hathras)\|(dalit)\|(rape) | 996 | 900 | 96 |
| Babri Accused Aquittal | Sep 30 2020 | substring:(?i)(babri) | 32 | 29 | 3 |
| Journalist Arrest | Oct 5 2020 | substring:(?i)(kappan)\|(siddique)\|(hathras)\|(dalit)\|(rape)\|(manishavalmiki)\|(manisha valmiki) | 466 | 433 | 33 |
| Bihar Manifesto | Oct 22 2020 | (?i)(?=.*bihar)((?=.*election)\|(?=.*manifesto)\|(?=.*vote)\|(?=.*vaccine)\|(?=.*bjp)\|(?=.*modi))\|((?=.*vote)\|(?=.*bjp)\|(?=.*modi))(?=.*vaccine) | 2455 | 2081 | 374 |
| Love Jihad Yogi | Oct 31 2020 | (lovejiha)\|(love jiha)\|(love-jiha) | 7 | 7 | 0 |
| Diwali | Nov 14 2020 | (diwali)\|(deepavali)\|(deepawali) | 1152 | 1074 | 78 |
| Unlawful Conversion Act (Love Jihad Law) | Nov 28 2020 | (lovejiha)\|(love jiha)\|(love-jiha) | 9 | 7 | 2 |
| Bengal Rally | Dec 12 2020-12 | (?i)(bengal)\|(rally)\|(rallies)\|(election) | 803 | 714 | 89 |

## B ORGANIZATION NAME KEYWORDS

**Table 4: Keywords used to filter out organization names from the tweeters after lower-casing the usernames**

group, team, organization, foundation, official, college, university, universities, fan, fc, school, institute, institutions, chamber, brand, service, board, bureau, gov, division, technology, consult, khabar, voice, collector, medical, health, mirror, journal, chronicle, post, daily, times, today, channel, temple, station, bjp, congress, council, business, shop, party, bollywood, cinema, academy, center, centre, state, collective, association, indian, group, sangh, NGO, RBI, online, cooperative, retail, .com, .in, .edu, .org, hospital, research, solution, department, bank, adani, fan, HSBC, sena, dpro, logic, tech, district, state, work, CPI, INC, BSP, AAP, CPM, NCP, BJP, trust, govt, Prakashan, corporation, socialist, communist, committee, janta

## C SELECTING MUSLIM CLASSIFICATION THRESHOLD

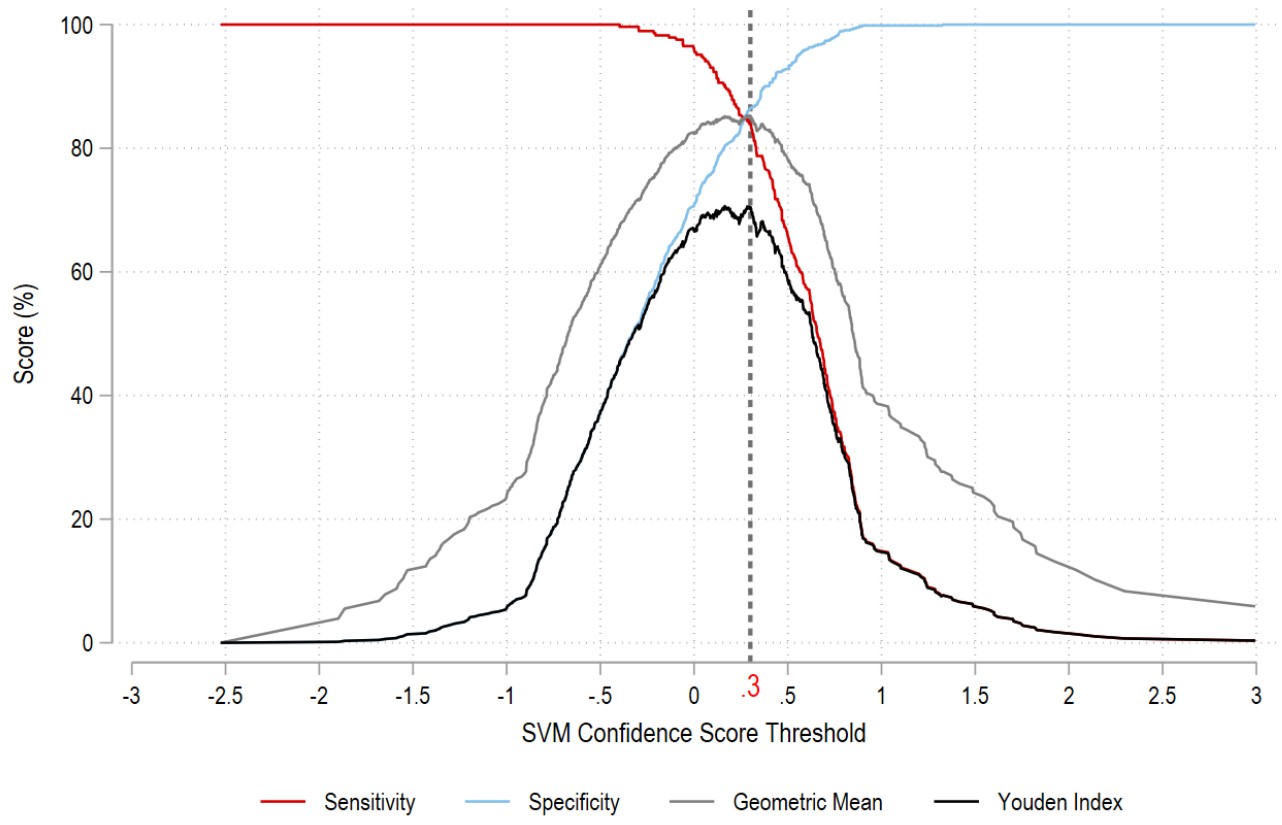

**Figure 5: Sensitivity, specificity, Youden index, and geometric mean by prediction threshold.**

## D STOPWORDS USED FOR COMPUTING BOW-GCS

appreciate, been, onto, mainly, wish, whence, with, yeh, also, certain, shes, cud, then, seemed, koi, that, several, causes, obviously, un, mightnt, nor, everything, value, ko, both, appropriate, thoroughly, within, youve, aur, than, of, eg, h, upon, theres, old, couldnt, ll, especially, p, e, presumably, c, et, edu, meanwhile, somewhere, regardless, trying, yours, perhaps, per, it, during, believe, itself, dont, forth, mustnt, tends, those, her, together, keeps, other, mein, urself, no, anyway, doesnt, who, known, shes, brief, themselves, have, provides, beforehand, sure, zero, km, whereby, enough, specified, here, toward, ss, etc, able, she, across, theirs, probably, qv, wherever, think, did, name, nine, according, aside, hi, containing, although, allows, hopefully, none, furthermore, secondly, o, because, want, usually, seriously, may, com, would, fifth, becomes, liked, r, re, oh, won, tweet, whether, whither, ain, try, ya, kept, is, through, different, given, hai, sir, howbeit, himself, already, placed, against, btw, beside, he, y, knows, ones, wasn, tried, shud, didnt, dis, likely, yes, else, inward, latterly, via, l, so, thru, inner, twice, nowhere, normally, dont, willing, ours, needn, what, do, hardly, man, nearly, less, behind, my, st, thats, haven, mustve, getting, hereupon, whom, not, indicated, shell, hers, afterwards, going, happens, shant, selves, hereafter, useful, shall, further, are, us, many, non, for, specify, regards, towards, nd, okay, goes, under, yourselves, beyond, will, done, their, uucp, herself, couldve, much, couldn, yu, most, inc, ourselves, u, saying, due, got, apart, throughout, where, help, anyhow, actually, into, i, reasonably, soon, either, some, more, nevertheless, thereupon, sorry, hain, k, herein, hereby, ab, six, two, below, z, tl, kuch, everywhere, n, novel, ji, downwards, t, unless, hell, hello, d, didnt, bt, ive, wants, anybody, ex, shouldve, why, among, anyone, dnt, hasn, about, thanx, relatively, him, nothing, whatever, still, youll, saw, ltd, ur, which, after, maybe, coz, best, sensible, now, cannot, aa, greetings, nhi, too, seen, in, tha, someone, shouldnt, around, being, elsewhere, them, wherein, yah, mightve, wouldve, m, corresponding, appear, shan, entirely, hither, though, instead, over, hasnt, everybody, away, consider, tweets, immediate, hadnt, sometimes, once, wonder, said, lo, way, currently, were, changes, wouldnt, near, to, you, needs, or, whereafter, really, hoga, four, know, couldnt, should, and, otherwise, out, far, whose, hes, don, has, comes, as, but, only, various, didn, following, specifying, its, doing, vs, sometime, hasnt, particularly, thereafter, shouldnt, therefore, bhai, seeing, wasnt, became, moreover, like, yet, theyre, insofar, contain, thanks, neednt, b, hadn, arent, say, must, somewhat, least, well, uses, consequently, latter, used, contains, somebody, que, co, third, always, off, such, use, ki, had, se, along, last, before, better, see, ho, twitter, second, available, however, self, went, mightn, does, guy, lest, kal, wont, again, huh, sub, doesn, by, inasmuch, truly, gives, might, youre, something, having, gaya, awfully, look, im, these, havent, keep, example, kr, eight, his, every, ie, next, theyve, if, get, whole, particular, lately, ignored, plus, ke, alone, a, could, rd, came, clearly, need, since, please, come, never, follows, later, th, therein, whenever, ve, toh, except, all, w, aren, doesnt, above, nobody, anywhere, wouldn, looks, ek, mostly, v, respectively, asking, neednt, somehow, mustn, s, youre, thorough, mustnt, cause, am, neither, your, considering, just, ok, same, little, havent, former, down, on, youd, exactly, indicates, cd, namely, followed, everyone, associated, until, an, up, kya, new, amongst, even, necessary, looking, allow, whereupon, was, sup, took, seven, youve, x, isnt, mean, overall, ever, werent, using, ought, nahi, without, thatll, merely, besides, jo, me, rather, ask, any, becoming, f, whereas, anything, le, how, definitely, own, while, this, unto, very, five, often, sent, let, despite, few, go, they, indicate, tell, one, our, there, unfortunately, almost, q, others, be, right, become, indeed, myself, formerly, wouldnt, cant, says, take, from, unlikely, yourself, shouldn, hence, whoever, course, each, na, welcome, gotten, isn, at, serious, thereby, anyways, ka, accordingly, outside, when, can, possible, seems, viz, between, its, another, thus, we, ohhh, thank, kar, the, quite, thence, gets, shouldve, bhi, first, tries, j, described, three, certainly, taken, regarding, weren, gone, noone, ye, g, seem, concerning, ma, seeming.

# E    BOW-GCS AND CONTEXTUALIZED-GCS VS. TWEET LENGTH

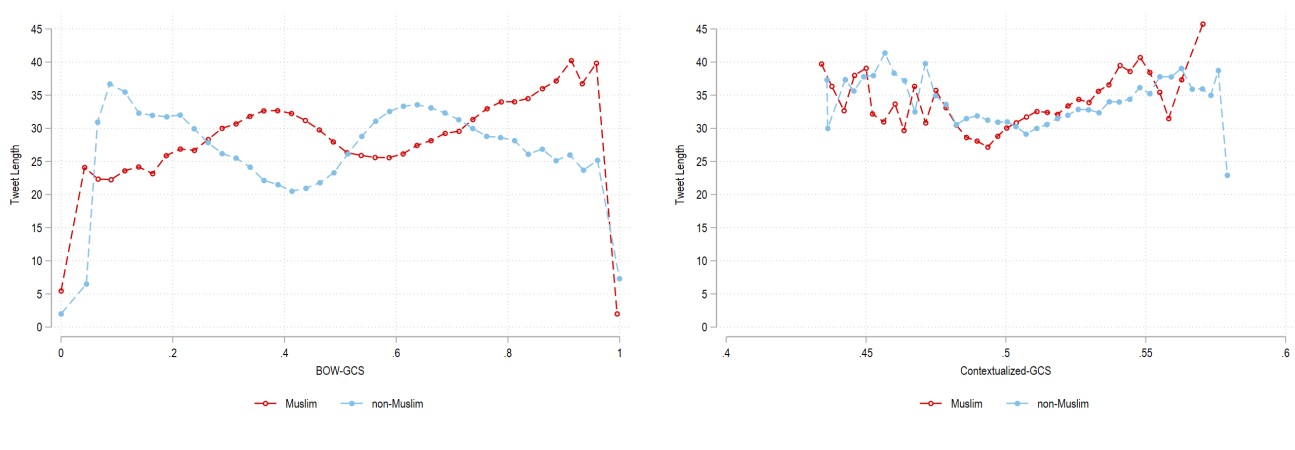

(a) BOW-GCS

(b) Contextualized-GCS

**Figure 6: Tweet length and GCS.**

# F COVID TWEETS TOPICS

**Table 5: Top 50 keyphrases for each topic based on c-TF-IDF score**

| Topic | Keywords/Phrases |
|---|---|
| **COVID Response** | virus, india, new_case, case, patient, positive_case, tested_positive, active_case, death, pandemic, day, positive, hospital, time, test, people, today, lockdown, fight, total_case, update, doctor, test_positive, testing, spread, vaccine, country, treatment, infection, delhi, infected, year_old, world, case_death, disease, state, need, report, warrior, social_distancing, case_reported, come, dr, outbreak, mumbai, indiafights, life, symptom, app, number |
| **Politics-Religion** | india, modi, bjp, govt, pm, people, indian, state, delhi, congress, cm, fight, muslim, lockdown, time, crisis, pm_modi, country, hindu, virus, party, indiafights, government, medium, politics, nation, leader, pakistan, pandemic, bihar, rahul_gandhi, maharashtra, shame, police, issue, modiji, minister, mp, student, economy, exam, kashmir, modi_govt, kejriwal, ppl, election, opposition, politician, situation, come |
| **China & Global** | china, wuhan, wuhan_virus, chinese, chinese_virus, virus, world, chinesevirus, beijing, china_virus, ccp, country, chinavirus, whole_world, wuhan_china, taiwan, wuhanvirus, wuhan_lab, usa, india, lab, china_wuhan, trump, virus_china, xi, xi_jinping, pandemic, virus_wuhan, chinaliedpeopledied, hong_kong, spread, entire_world, china_must, outbreak, nation, pakistan, italy, war, biological_weapon, communist, rest_world, wuhan_health, wet_market, time, started, chinese_govt, china_china, chinese_wuhan, origin, economy |
| **Socio-Economic** | lockdown, time, india, crisis, pandemic, business, work, market, company, day, bank, post, year, fund, today, help, economy, need, industry, demand, virus, thanks, service, month, home, job, people, money, good, situation, customer, online, support, employee, student, loan, thank, issue, impact, app, family, march, read, order, sale, pm, life, sector, state, hope |

**Table 6: Sample Tweets from each Topic Category from both religious groups**

| Topic | Tweets |
|---|---|
| COVID Response | (1) PLEASE SHARE - A list of all #COVID19 Helpline numbers from various States/UTs #CoronaVirus #COVID2019 #Corona #CoronaAlert #COVID19 #HelpUsToHelpYou <url>
(2) #SerumInstitute denies reports Indians to get free shots of #COVID19vaccine in 73 days #COVID19 #Coronavirus #News #India
(3) @realDonaldTrump I would like to suggest use of Traditional Medicines like Unani and Ayurveda. Which is more effective in treating such diseases or viruses. Please start using it and save life's of human beings. We are praying for whole humanity. Very soon Corona will be gone by the grace of LORD
(4) #Nationfirst Inshallah We will fight against Corona with unity.
(5) Sir, <person-name>-23 suspect Covid-19 wants a bed in any Hospital, they have visited 7 Hospitals No beds available,Pl arrange a bed in any hospital & call <cellphone>. |
| Politics-Religion | (1) @ApteVLA @DelhiPolice @CPDelhi @DCPSouthDelhi @drharshvardhan More outrageous is how was a congregation of 1000+ people was being held by Muslims in the heart of Delhi i.e. Nizamuddin & Delhi Police really had no clue about it. Now just praying a Corona explosion does not take place of this callousness. @AmitShah @PIBHomeAffairs @PMOIndia
(2) Right now bjp is very strong and able to win easily 20+ seats out of 25 but if CM fail to handle #Corona ,migrant labour issues ,current medical anarchy ,and solely depending on beurocrates he will loose badly .One more thing Cm is popularity is decreasing day by day .
(3) What Demonetization, Aadhaar, NPR & CAA did not allow CoVID-19 will do for @narendramodi Unite India Black money, Poverty, anti-Hindutva agenda are all pale Villain Mahamari i.e epidemic is an enemy worth rallying against
(4) @OmarAbdullah seeing what your grandfather become so loyal to India? Where did he leave us. This countries PM instead of combating corona virus is asking people thali bajaav & now diya jalaav. Like what made him not to listen to Jinnah. Can you please give Farooq Abdullah's no.?
(5) @ShashiTharoor Some of the fake news which can be taken care by Congress in its support base 1. If a health worker asks number of family members, its not for NRC so kindly help him. 2. If any corona patient from muslim area is taken for treatment, he will not be killed by injection. |
| China & Global | (1) Even assuming the outbreak was an accident of nature the fact that it didn't travel beyond #Wuhan but went global shows deliberate hiding of facts &a mishandling by #China. Given the virus is not natural China is again complicit in this global tragedy. The rest is academic
(2) @KamalaHarris No ma'am actually it is universal disaster caused by China virus #Trump is working hard to check this china virus (covid-19) #Trump taken all prication to save peoples life #America should move away from China's orbit
(3) Communist OLI running his government on communist Chinese OIL. Nepal thinks Chinese could save them from India like Pakistan was thinking days before but they don't know china is now destabilized after Chinese Wuhan virus. #EnoughIsEnough
(4) **C**-hina **O**-riginated **V**-iral **I**-nfectious **D**-isease #COVID #WuhanCoronavius
(5) Whole #World knows mortality rate is 10-15#Europe #US spread data shows what really happened in #China @JackMa saying we #manufacturing #Ventilators first #Chinese produced #Goods to Live then for #Death #DeathToll will be #Millions by June2020 |
| Socio-Economic | (1) @realDonaldTrump What I admire most about you is how secure you must feel to post scenes from your time-wasting Covid-trapping rallies whilst you're busy spending taxpayer resources on not doing your job.
(2) All the kids who wrote JEE Mains (B Arch papers) today Sept 1st, 2020 would have been under tremendous pressure. Parents will be in constant pain and pressure. COVID Situation apart, the real admission and classes (on campus) would be the next challenge. Hope things work out well
(3) Indian Banks' Association (IBA) is finalising a special restructuring proposal seeking for a year long moratorium for impacted companies. Association feels it will take at least a year for recovery #CoronaCrisis #EconomicCrisis
(4) What will happen when the news channels start discussing GDP, Economic crisis, Covid, lockdown, depression?? Will anything change by their reporting? It didn't change anything when they used to do before June 14th BUT (1/2)
(5) Sir oil price nosedived/but people didn't got the benifit of the low crude price #people expecting fall in petrol/desil price at pumps after Corona lockdown #will govt give concession or make good the low crude price saving to Corona expenditure? |

# G HIGH GCS TWEET EXAMPLES POST EVENTS

## G.1 Embedding-GCS

**Table 7: High Contextualized-GCS examples sampled 7 day post-event. A summary is provided before the example tweets.**

| Event | Non Muslims | Muslims |
|---|---|---|
| Janata Curfew | China Blame, welcome Janata Curfew, bolster frontline workers
(1) #user participated and salutes all the Doctors, Nurses, health workers, sanitary workers, media and police who are working & fighting against #Corona. #JaiHind #JanataCurfew #CoronavirusPandemic #Covid_19india
(2) China - Manufacturer & Exporter of Corona Virus across the Globe and responsible for the sufferings of millions. #ChineseVirus
(3) They shd be charged with bioterrorism #CoronaJihad #coronaterrorists #Corona #coronavirusindia #ChineseWuhanVirus #ChinaVirus #WuhanCoronavius #Wuhan
(4) Public is not threatened but those in Admin need to be made aware of the seriousness of the situation. Hence DM Act. Response from States were in fits and starts till PM himself imposed janta Curfew. No one imagined the tremendous response frm public. Now onus is on babuz.#corona
(5) CHINA must be dragged into International Court and stripped of its VETO power in the UN 'Crime against humanity' COVID-19 is a Chinese Virus. Copy and paste. #ChineseVirus | Predominantly Kashmir news (the only Muslim majority state in India), encouraging Lockdown; World news
(1) #BreakingNews: #Jordon reports first #Covid_19 death. According to Petra, 83 year old Women had suffered from blood poisoning. #COVID2019 #CoronavirusLockdown
(2) #COVID-19 Today's picture from Lal Chowk Srinagar. Complete Lockdown. Lockdown in Kashmir is a routine, nothing new. Difference is that this time #COVID-19 is responsible for lockdown.
(3) #Flash Pakistan cricketers to donate Rs 5 million to govt emergency fund for Covid-19 pandemic..... #PakistanVsCorona #Pakistanis #PakistanFightsCorona'
(4) Prophet Muhammad (SAW) said, If you hear of an outbreak of plague in a land, do not enter it; but if the plague breaks out in a place while you are in it, do not leave that place. how to prevent Covid-19/CoronaVirus. He said this approximately 1400 yrs ago
(5) 7 more test positive in Kashmir, cases surge to 27 in J&K #COVID_19 #StayAtHomeStaySafe |
| Tablighi | Show of unity in fight against corona
(1) Today #9pm9minute Lets light up for a brighter future and spread hope & motivation to all humanity in this fight against Corona. Spread the light of unity, spread the light of Safety. Stay Home Stay Safe. We are ready ji. INDIA
(2) Thanks all the Committed Team memers of BJYM,CENTRAL DIST & IMPHAL WEST Dist. for your dedication to Combat Covid-19. We appreciate for achieving our target.#FeedTheNeedy 3000+ in 6 hours.Great appreciation ji.
(3) Turn off all d lights of ur home. Stand at your doors or balconies. Light diyas or torches for 9 min. Do not cross d Lakshman Rekha of Social Distancing.Challenge d darkness spread by Corona crisis & introduce it to the power of light #AaoPhirSeDiyaJalayen
(4) The message is loud and clear.. 130 crore Indians and are with PM Shri and PM Shri ji is with 130 crore Indians.. Together let's fight against Corona.. #9pm9minute
(5) Show your unity and throw away corona. Because there is power in Unity. #Lightforglory #IndiaFightsCorona #Indiaglows #lightdiya9pm9mins | Criticizing Indian media for Islamophobia
(1) Sooner or later our nation will find cure from #COVID but #Islamophobia like all other forms of racism has remained an incurable disease & Indian media is a big contagion of this disease
(2) Dear Indian Media, What about the 70-year-old superspreader guru, Baldev Singh? Warm regards, Indian. #COVID #COVID19 #COVID-19 #Covid_19india #coronavirus #Coronavirustruth #CoronavirusOutbreak #CoronaVirusUpdates #CoronaVirusUpdate
(3) India has more big social virus than Corona! #CommunalVirus #CasteVirus #HealthForAll #Nizamuddin #TablighiJamat
(4) What happened in Nizamudin Markaz presently is being called as the Corona Jihad. They need villains to put the entire blame of the pandemic and hide the failure of the government in containing the spread of covid19. #NizamuddinMarkaz #Islamophobia
(5) Reality of #COVID Born in China Grown up in Italy Become mature in America Become Muslim in India #mediavirus #foolishmedia |

**Table 7: High Contextualized-GCS examples sampled 7 day post-event. A summary is provided before the example tweets.**

| Event | Non Muslims | Muslims |
|---|---|---|
| Migrant Deaths | COVID related news. Some mention of labour suffering

(1) corona virus in slaughterhouses: politicians call for stricter controls
(2) the labour in this country have suffered miserably due to successive governments buckling down to left pressure and continuing with archaic labour laws. in the bargain they stifled industry. china prospered because they respected industry. time we get this right post corona
(3) one trusts corona figures coming from west bengal, as my family lives there, they says very pathetic condition, tmc goons charging hectic amount in hospital.
(4) end 'tsunami of hate and xenophobia' sparked by covid-19, says un chief. migrants & refugees being vilified as a source of virus in denying medical treatment
(5) the economy is in free fall. so why isn't the stock market?
(6) the continuous inmigration and outmigration of stranded labourers going on in different parts of india may affect the economy of all these concerned states. it will also change unemployment figures.their safety must be ensured from the corona virus and its further spreading also | Sharing the incidences of increasing islamophobia

(1) an audio clip of maulana saad, in which he insisted that #covid_19 can do no harm to muslims and asked jamaat members not to follow social distancing norms, is possibly "doctored" and fake. this audio circulated widely by indian media. #muslimphobia_in_india
(2) we turned masjids, madrasa hostel into quarantine centers , we donated plasma for other patients ,we donated blood, we donated to #pmcaresfund and still we are blamed by terrorist organization like headed by terror chief
(3) the shocking news of #aurangabad proves the poors and laborers lives in our country is not a concern to gov't, they only care about the rich. may almighty protect & safeguard the health of migrants worker and poors in this pandemic #covid_19 #trainaccident #migrantlivesmatter
(4) where is the money donated by the people of india being spent...? #covid_19 #pmcaresfund
(5) #indiaisnotwithzeenews when news channel supposed to show news about covid-19, migrant workers, govt relief toward this, condition of health facilities, talk with doctors etc.. instead of this they are showing rediculous content like jihad, hindu muslim, debate wid spoksprson etc |
| Coronil Launch | Pride and relief at launch of Coronil Ayurvedic remedy, little skepticism in some tweets.

(1) #coronil during the launch, #patanjali co-founder #babaramdev said that their clinical trial found that 69% of the patients tested recovered from covid-19 within three days, while 100% of the patients recovered within a week
(2) coronavirus vaccine update: #patanjali launches coronil drug, claims drug can cure #corona in 14 days #coronavirus #covid19
(3) first corona medicine prepared by patanjali. <3 clinical trials done , medicine launched today.. 69% patient recovered in 3 days 100% recovered in 7 days. #patanjaliayurved
(4) watch 's broadcast: join us for a live session with eminent doctors of india, who will give answers to all your questions about covid-19. #indiafightscorona #stayhome
(5) #verified ayush ministry asks ramdev to tell composition of medicine being claimed for covid; site/hospital where the research study was conducted; protocol, sample size, ethics panel clearance & ctri registration patanjali asked to stop advertising such claims immediately | Calling out discriminatory media reports calling Tablighi a super-spreader event and not discussing approval of Hindu religious congregation by government and courts. Also, against exam and Coronil, calling it a fake drug.

(1) during tablighi jamat congregation many other religious congregation happened. none was called corona bombs & terrorists but on tablighi and all muslims are facing discrimination upto now though that was the starting phase of pandemic. now yatra is allowed in worst corona time
(2) salute to shahnawaz shaikh. it is observed,for last four months,muslims are in front line to help the needy. more power to corona warriors.almighty ast protect you from evil forces.ameen....
(3) thank god for the majoritarian privileges. no one will bat an eyelid on jagannath rath yath. with more than 4.4 lakh nationwide covid-19 cases, it still has the blessings of the center, the state, the other states with no links to it #thisthread
(4) urge you to arrest patanjali for spreading fake corona drugs during covid times.
(5) lockdowns failed to control the spread of covid 19 and now government of karnataka is planning to spread the virus more intensively by conducting sslc exams and risking lives of lakhs of students. #postponesslcexam |

**Table 7: High Contextualized-GCS examples sampled 7 day post-event. A summary is provided before the example tweets.**

| Event | Non Muslims | Muslims |
|---|---|---|
| Exam Satyagraha | COVID News and also against in person exams during COVID
(1) covid-19 vaccine won't end pandemic alone: who warns
(2) & rest all central & state government pls arrange bus,train services for all students who are attending jee & neet exams & facing problems in reaching out their venue. and pls arrange new dates for those aspirants who are covid patient
(3) in this exam center if you positive for corona..nobody cannot save you and your family....even they don't give money for the treatment of corona
(4) covid-19: tamil nadu reports 5,980 new infection cases, 80 fatalities
(5) there is nothing as of now for the covid +ve candidates, for mains itself, how can we expect for adv. also sir mr venugopal rao before the judgement of sc himself said that there is no sky falling on us if exams are not conducted, but aftr the judgement he has not made any remark | Harsh criticism of government for conducting in person exams during COVID
(1) while modi government is afraid to call parliament session to discuss covid and pm cares fund, they are asking students to attend exams! #satyagrah_againstexamsincovid #studentskemannkibatt #rahulgandhi
(2) i appeal to the government again, to postpone exams. covid is no joke and exposure could lead to life long complications. even a single life lost is not worth it. in these strange times, surely students shouldn't have to risk their life for an exam. #satyagrah_againstexamsincovid
(3) for the modi govt, college management lobby and paid tuition mafia is more important than the lives and careers of 50 lakh students. what else explains it's insistence on conducting jee and nee now during peak-covid? #studentskemannkibaat #satyagrah_againstexamsincovid
(4) the growing opposition against jee neet exams is only here.many major media's of the respective states have remained silent about it.they will report it when the first student who wrote the exam tests +ve for covid. #satyagrah_againstexamsincovid
(5) we want postponement of all the exams in september covid is not a joke #nationwantsjee_neetpostponement #webelievein_swamyji |
| GDP Contraction | World News, COVID medicine and vaccine
(1) Reasons for the day-trading trend include people stuck at home who are bored and/or trying to replace lost income, the stock market's recent volatility, and a reduced number of sporting events available for online sports betting. #trending
(2) German minister spat at and verbally abused at Covid protest Dirty behavior
(3) HealthPartners recruiting Minnesotans for COVID-19 vaccine trial
(4) FAQ's on #COVID_19 Management from AIIMS, New Delhi E-ICU 1. Tocilizumab: Limited role, experimental 2. Favipiravir: Not Recommended 3.Remdesivir: Not on suspected cases, but recommended for 5 days, OD. 4. Ivermectin: Not recommended 5.Plasma Therapy: Use with caution.
(5) There is a theory that Corona has lost its potency. But what if the low fatality is a function of the rapid tests that show even low viral loads as positive. This means the virus might be as deadly as before. So maybe we should continue to be cautious. | Criticizing PM for GDP decline, unemployment, student protests, border clashes, etc.
(1) Non Congress Prime Minister with long duration, Supreme Court Judges came to pressmeet, Prime Minister to tour foreign countries, most Buisness mans loan waved off, China in laddak PM in Photoshop History of Corona 37 lacs, GDP is in top most in the world -24%, #sabchangacee
(2) India is reeling under Modi-made disasters: Historic GDP reduction -23.9% Highest Unemployment in 45 yrs 12 Crs job loss Centre not paying States their GST dues Globally highest COVID-19 daily cases and deaths External aggression at our borders #RahulGandhiSpeaksOnEconomy
(3) India is reeling under Modi-made disasters: 1. Historic GDP reduction -23.9% 2. Highest Unemployment in 45 yrs 3. 12 Crs job loss 4. Centre not paying States their GST dues 5. Globally highest COVID-19 daily cases and deaths 6. External aggression at our borders #GDPTruth
(4) India is reeling under Modi-made disasters: 1. Historic GDP reduction -23.9% 2. Highest Unemployment in 45 yrs 3. 12 Crs job loss 4. Centre not paying States their GST dues 5. Globally highest COVID-19 daily cases and deaths Bure din hi wapas de do bhai
(5) India's GDP at -23.9,covid cases daily reporting 70k+,flood crisis in states,JEE-NEET in pandemic,SSC students protesting Toh mera sawal hai : Aaj nashte mei Rhea ne kya khaya hoga? #SpeakUpforSSCRailwaysStudends #JEE-NEET #GDPDrop #COVID19 #mediascum #MediaBias |

**Table 7: High Contextualized-GCS examples sampled 7 day post-event. A summary is provided before the example tweets.**

| Event | Non Muslims | Muslims |
|---|---|---|
| Bihar Manifesto | COVID related news 
 (1) Addressing the inaugural function of the grand challenges annual meeting 2020, pm modi said, ...india is now at the forefront of vaccine development for covid-19. #trending 
 (2) media advisory - who africa online press briefing on rapid diagnostic tests for covid-19 in africa – 22 october 
 (3) bjp's covid-19 vaccine promise in bihar causes stir; oxford says trial to continue after death of volunteer via 
 (4) the bengali community in the delhi-ncr region is staying away from durga puja festivities, a community affair in parks and other open spaces. and nowhere is this more evident than in chittaranjan park, often known as 'mini bengal'.covid-19: durga puja ce… 
 (5) stage set for country's first covid-era elections in bihar \| india news | Strong condemnation of Bihar Manifesto of ruling party promising vaccine for vote 
 (1) brings dirty politics to their #bihar manifesto: "vaccine for vote" . so only #bihar gets vaccine free & rest of india pays? does this mean if doesn't win then won't people of bihar get #corona vaccine?politicising a pandemic is utter national shame! 
 (2) #biharelections2020 #covid19 bjp assures to give corona vaccines if they wins bihar election 
 (3) bjp in bihar election :free covid vaccines for all ! *only for bihar,the rest of the country don't matter rn. * also only if you vote for us,will give you this non-existent yet vaccine *dosen't matter that we're currently leading country & should provide it to all 
 (4) congress: free polio vaccines to all indians bjp-rss: we will provide free covid vaccine to states having elections (bihar) if you give us votes!! 
 (5) the free vaccination promise by bjp for bihar election manifesto is illegal and unlawful. this renders opposition at disadvantage. the vaccine program for covid-like pandemic is a universal immunization process and not just state specific which can be used for electoral benefit. |

## G.2 High BOW-GCS Tweets

**Table 8: High BOW-based GCS examples sampled 7 day post-event. For long tweets, a summary is provided before tweet examples.**

| Event | Non Muslims | Muslims |
|---|---|---|
| Janata Curfew | (1) Chup be 
 (2) Whistle 
 (3) Yes it is down. Here is the distant next best 
 (4) 
 #SantRampalJi_CanEndCorona Please 
 (5) Thanks yar | COVID spread and lockdown news 
 (1) The funeral of the 65-year-old businessman, the first person in Kashmir to succumb to coronavirus (COVID-19), the disease that has swept across the globe, took place on Thursday in north Kashmir's Sopore town of Baramulla district. 
 (2) #BigBreaking 7 More Test Positive In Kashmir, COVID-19 Cases Surge To 27 In Kashmir 
 (3) #COVID-19 Closure of all religious places in Srinagar is underway with active cooperation of Managenent committees. Revered shrines Hazratbal, Naqshband Saheb show the way: DC Srinagar 
 (4) May Allah have Mercy on us #covid_19_kashmir 
 (5) 5 More Test Positive In Kashmir, COVID-19 Cases Surge To 33 In J&K. 2 from kashmir and 3 from Jammu. #BREAKING #COVID2019 #JammuAndKashmir |

**Table 8: High BOW-based GCS examples sampled 7 day post-event. For long tweets, a summary is provided before tweet examples.**

| Event | Non Muslims | Muslims |
|---|---|---|
| Tablighi | (1) bash on regardless 
 (2) Yess Sir !!!please Help us 
 (3) Woah 
 (4) Are they retarded ?? 
 (5) Heartbreaking | COVID relief by Muslim leaders, taking back domicile law in Kashmir that might change demographics of Muslim majority state, calls for releasing student activist arrested for inciting communal violence during Delhi riots. 
 (1) Habeeb- E- Millat Janab Akbaruddin Owaisi Sahab Inspecting The Ration Kits Wich Are Been Distributing To Needy people In Hyderabad. #akbaruddinowaisi #covid #helpingothers #alhmdulillah 
 (2) #DomicileLaw is dangerous than #Covid_19 for J&K Youth, no more destruction of Youth. Youth Maange Apna Haq Outsiders are not allowed One voice, one Slogan. #rollbackdomicilelaw #rollbackdomicilelaw #rollbackdomicilelaw #rollbackdomicilelaw #rollbackdomicilelaw 
 (3) Instead of 'Corona Hunting' 'Witch Hunting' is going on #releasemeeranhaider 
 (4) Relief (Ration kits) was distributed in MCH colony & meer sagar kunta in Jahanuma division by Bahadurpura MLA among the people who have been affected due to #Covid_19 lockdown AIMIM Corporator Hussaini Pasha overseen the distribution. 
 (5) The virus of hate derived from Media Virus is more Dangerous than corona So the Best thing to do now is Stop watching TV, especially News channels |
| Migrant Deaths | (1) i agree with you, ... 
 (2) deleted? 
 (3) anyone else teared up a bit with pride :') 
 (4) #uttarpradesh for sure 
 (5) civilizational ethos! way to go. | About increasing islamophobia in India 
 (1) break the chains of hate undo the inhumanacts #skssf #uapa #skssfhomeprotest #safoora_zargar #protestlockdown #email #covid_19 
 (2) triple talaq —– target muslim article 370 —–target muslim caa nrc npr — target muslim delhi 'genocide' —- target muslim jamia students — target muslim corona pandemic — target muslim #muslimphobia_in_india 
 (3) #muslimphobia_in_india triple talaq target muslim article 370 target muslim caa nrc npr target muslim delhi 'genocide' target muslim jamia students target muslim corona pandemic target muslim #muslimphobia_in_india 
 (4) #muslimphobia_in_india lynching of muslims in the name of beard. lynching of muslims in the name of cap. lynching of muslims in the name of cow. lynching of muslims in the name of j s r. lynching of muslims in the name of corona. when will we bleed? #muslimphobia_in_india 
 (5) #covid-19 54 persons tested positive in kashmir valley since yesterday evening. 28 kulgam district. 10 handwara. 07 from jammu division. 05 from anantnag. 02 from reasi. 01 from kathua. 01 shopian. out of 54 positive samples, 47 tested positive at skims and 07 at cd hosptl |
| Coronil Launch | (1) indeed. and that too for a while q 3/4 of fy 2017 !!! 
 (2) use the pause 
 (3) screenshot from 
 (4) seems this surgeon will unveil mystery 
 (5) #oneindia. # beindianbuyindian | Calling out hypocrisy in congregations at COVID times and other COVID-related news 
 (1) people who branded covid 'muslim virus' after tablighi jamaat are quite on odisha's rath yatra! and the rath yatra was sanctioned by the supreme court at the height of india's covid spike! it's all in the name as usual... 
 (2) seven more pakistan players test positive for covid-19 - (1)kashif bhatti (2)mohammad hasnain (3)fakhar zaman (4)mohammad rizwan (5)mohammad hafeez (6)wahab riaz (7)imran khan..... 
 (3) 45-year-old woman from south kashmir dies of covid-19, j&k toll 93 
 (4) #namastetrump welcome ceremony of corona 
 (5) earthquakes from the earth, locusts down from the sky & the virus like corona between earth and the sky are testifying to the fact that the creator of all universe is angry with us. it's a time to repent. #timetorepent |

**Table 8: High BOW-based GCS examples sampled 7 day post-event. For long tweets, a summary is provided before tweet examples.**

| Event | Non Muslims | Muslims |
|---|---|---|
| Exam Satyagraha | (1) this is sobering. the unraveling of america
(2) at his usual analytical best ..
(3) how reopening multiplexes and hotels may pan out?
(4) impressive #bengalfightscorona
(5) wonderful. i admire. | Against long imprisonment of muslim student leader condemning center and state governments for spreading hate against muslims
(1) history will remember corona less as pandemic & more a drive of witch-hunt & persecution of muslims. this pandemic has proven to be powerful tool for fascist forces to slay #muslim leading voices. #releaseasiftanha #100daysofinjustice
(2) we are suffering, not just from covid, but crisis of hunger and crisis of hatred against #muslims. state is racially profiling anti caa protesters we will to do what is the duty of citizens. you cannot silence the truth by jailing us. #releaseasiftanha #100daysofinjustice
(3) #aimim chief barrister asaduddin owaisi representation letter to cm kcr on consideration for the exempting of motor vehicle tax during the period of covid-19 pandemic lock down
(4) muslims are abused for clearing #upsc. muslims are arrested for protesting against biased laws like caa. muslims are falsely accused of spreading corona. muslims are lynched by mobs without repercussion false propaganda by govt #sabkasathsabkavikash ? #upsc_jihad
(5) kejriwal was more sanghi than sanghis. to blame jamaatis for covid, he repeatedly and explicitly blamed them. he also gave break up of infections split between jamaati related and non jamaatis.#kejriwal_must_apologize'] |
| GDP Contraction | (1) fyi please
(2) -19 Identifies The Significance Of Rewards And Wellness Programs on
(3) This is huge
(4) Sorry the above tweet was sent by mistake.
(5) Neti(nasal cleansing) is even better | COVID relief shortcomings, COVID news
(1) As per the inclusion assessment of COVID-19 entitlements, 63% of the Dalit (out of 19,590) and 62% Adivasi (out of 2030) households, respectively, were not enrolled with the Ujjwala Yojana. #IfWeDoNotRise
(2) All residents within radius of 50-meter micro containment zones to be tested for COVID-19 in Sgr
(3) Intractable.......... The current banking challenge is the most intractable one even before COVID-19
(4) #NHMMP_Employee_wants_JusticeMAMA Many employees in contract health were corrected by Corona but the policy has not been implemented till June 5, 2018, such a selfish government has no right to live.
(5) Iam very thankful to Voulanteers of popular Front of India & SDPI, Kolar DISTRICT. Who participated in the last rites of a Christian brother Covid-19 in KGF taluk today. Till today voulanteers of PFI & SDPI of Kolar District completed respectful burial of 50 Covid-19 bodies. |

**Table 8: High BOW-based GCS examples sampled 7 day post-event. For long tweets, a summary is provided before tweet examples.**

| Event | Non Muslims | Muslims |
|---|---|---|
| Bihar Manifesto | (1) thank you mam
(2) does this look like an appropriate frother-seine interaction to you?
(3) thank you, roadside golgappas, samosas, and momos. :p
(4) a heartening reminder that we're more than our entrenched hatreds.
(5) now trump's gonna want a large consignment of filth. | Against French ban on Muslim long robes in schools, and sarcasm against Bihar's manifesto's vaccine for vote promise.
(1) approximately 28% of the world population is muslim. approximately $80 trillion dollars is the world gdp. if all muslim countries #boycott_french_products i can guarantee you that france would be on its knees within 6 months. french economy is already weak due to covid-19
(2) covid 19 shall engulf france because of its moron president macron... people will bear the fury of the disease on account for their bigoted president #boycottfranceproducts #shameonyoumacron #macronthedevil
(3) india's free covid vaccination calendar bihar: nov 2020 assam, kerala, tn, wb: apr 2021 goa, up, hp, uk: feb 2022 gujarat: dec 2022 karnataka: may 2023 mp, ch, rj: dec 2023 dates matching the schedule of assembly elections in the states are purely coincidental.
(4) urgently need of blood plasma for covid patient. name : syed ghulam mohi udin address : syed pora bata pora blood group: b postive admitted in skims soura. bed no 5 ward no. surgical observation if anybody is interested contact <tel.no.>
(5) people of bihar are going to vote on issues like unemployment, corruption, mismanagement of covid, women security, floods, farmer issue, which clearly signals the exit of bjp jdu govt from power #BiharWantsChange |

# H SUMMARY STATISTICS

Table 9: Descriptive Statistics for the dataset. Averages over event specific data are reported along with standard deviations in parentheses. *M: Muslim, NM: Non Muslim

| | EVENT | Janata Curfew | Tablighi | Migrant Deaths | Coronil Launch | Exam Satyagraha | GDP Contraction | Bihar Manifesto |
|---|---|---|---|---|---|---|---|---|
| | Date | Mar 22 2020 | Mar 31 2020 | May 8 2020 | Jun 23 2020 | Aug 23 2020 | Aug 31 2020 | Oct 22 2020 |
| | Muslim % | 8.33 | 9.11 | 7.86 | 7.81 | 8.38 | 8.07 | 6.54 |
| Interact | Overall% | 10.30 | 12.14 | 15.06 | 16.96 | 18.84 | 19.09 | 20.21 |
| | Muslim % | 54.34 | 53.08 | 62.55 | 67.04 | 65.87 | 69.61 | 75.38 |
| | Non Muslim % | 6.26 | 8.03 | 11.01 | 12.72 | 14.54 | 14.66 | 16.35 |
| GCS | Overall | 0.5007 | 0.5008 | 0.5014 | 0.5012 | 0.5007 | 0.5007 | 0.5009 |
| | | (0.002) | (0.0027) | (0.0026) | (0.0023) | (0.0024) | (0.0022) | (0.0024) |
| | *M Interact | 0.4997 | 0.5003 | 0.5005 | 0.4996 | 0.5002 | 0.4999 | 0.4994 |
| | | (0.0021) | (0.0028) | (0.0035) | (0.0025) | (0.0031) | (0.0025) | (0.0022) |
| | *NM Interact | 0.5006 | 0.5005 | 0.5008 | 0.5016 | 0.5009 | 0.5009 | 0.5009 |
| | | (0.0022) | (0.0027) | (0.0027) | (0.0021) | (0.0021) | (0.0021) | (0.002) |
| | *M non-Interact | 0.5007 | 0.5019 | 0.5018 | 0.501 | 0.5011 | 0.5007 | 0.5009 |
| | | (0.0022) | (0.0034) | (0.0033) | (0.0028) | (0.0033) | (0.003) | (0.0024) |
| | *NM non-Interact | 0.5007 | 0.5007 | 0.5015 | 0.5012 | 0.5007 | 0.5008 | 0.5009 |
| | | (0.002) | (0.0026) | (0.0025) | (0.0023) | (0.0023) | (0.0022) | (0.0024) |
| Topics | COVID Response | 0.69 | 0.64 | 0.59 | 0.58 | 0.59 | 0.57 | 0.59 |
| | | (0.27) | (0.22) | (0.19) | (0.19) | (0.2) | (0.19) | (0.19) |
| | Politics-Religion | 0.14 | 0.16 | 0.21 | 0.19 | 0.22 | 0.24 | 0.2 |
| | | (0.22) | (0.2) | (0.22) | (0.23) | (0.27) | (0.29) | (0.28) |
| | China & Global | 0.06 | 0.07 | 0.04 | 0.07 | 0.02 | 0.02 | 0.03 |
| | | (0.28) | (0.22) | (0.19) | (0.16) | (0.18) | (0.18) | (0.16) |
| | Socio-Economic | 0.11 | 0.13 | 0.17 | 0.15 | 0.16 | 0.16 | 0.17 |
| | | (0.16) | (0.17) | (0.2) | (0.23) | (0.25) | (0.25) | (0.24) |
| Emotions | Valence | 0.45 | 0.46 | 0.47 | 0.46 | 0.46 | 0.46 | 0.47 |
| | | (0.05) | (0.05) | (0.05) | (0.05) | (0.06) | (0.06) | (0.06) |
| | Fear | 0.45 | 0.45 | 0.44 | 0.45 | 0.45 | 0.45 | 0.44 |
| | | (0.06) | (0.05) | (0.05) | (0.05) | (0.05) | (0.05) | (0.06) |
| | Sadness | 0.41 | 0.42 | 0.41 | 0.41 | 0.42 | 0.42 | 0.41 |
| | | (0.04) | (0.04) | (0.04) | (0.04) | (0.05) | (0.05) | (0.05) |
| | Joy | 0.3 | 0.3 | 0.31 | 0.31 | 0.3 | 0.3 | 0.31 |
| | | (0.05) | (0.05) | (0.05) | (0.05) | (0.06) | (0.06) | (0.06) |
| | Anger | 0.44 | 0.44 | 0.44 | 0.44 | 0.44 | 0.44 | 0.43 |
| | | (0.05) | (0.04) | (0.04) | (0.05) | (0.05) | (0.05) | (0.05) |
| Ego-Net | Followers | 2496.63 | 2401.09 | 2867.94 | 3315.16 | 2954.83 | 2766.27 | 3823.92 |
| | | (8586.73) | (8384.17) | (9929.8) | (12296.33) | (10296.9) | (9375.01) | (11418.67) |
| | Friends | 936.18 | 935.06 | 988.95 | 1004.55 | 986.22 | 966.25 | 1137.37 |
| | | (1417.11) | (1439.22) | (1653.84) | (1532.91) | (1519.08) | (1431.95) | (1766.76) |
| Engagement | retweets | 1.93 | 2.27 | 2.41 | 2.57 | 3.51 | 4.28 | 2.87 |
| | | (19.35) | (16.01) | (16.4) | (16.28) | (22.88) | (38.91) | (15.4) |
| | Fraction of replies | 0.66 | 0.61 | 0.53 | 0.51 | 0.51 | 0.54 | 0.48 |
| | | (0.31) | (0.31) | (0.35) | (0.37) | (0.38) | (0.39) | (0.39) |
| | Tweet frequency | 7.5 | 12 | 16.07 | 13.08 | 10.73 | 9.89 | 9.34 |
| | | (10.08) | (16.93) | (25.58) | (24.94) | (19.72) | (19.96) | (17.6) |
| | Account days | 2493.81 | 2476.94 | 2490.63 | 2539.8 | 2465.07 | 2524.91 | 2635.2 |
| | | (1253.79) | (1244.03) | (1266.25) | (1292.43) | (1353.38) | (1335.86) | (1335.84) |
| | Tweeters | 4671 | 6946 | 6387 | 4622 | 3497 | 3792 | 1989 |

# I  TREATMENT EFFECT HETEROGENEITY

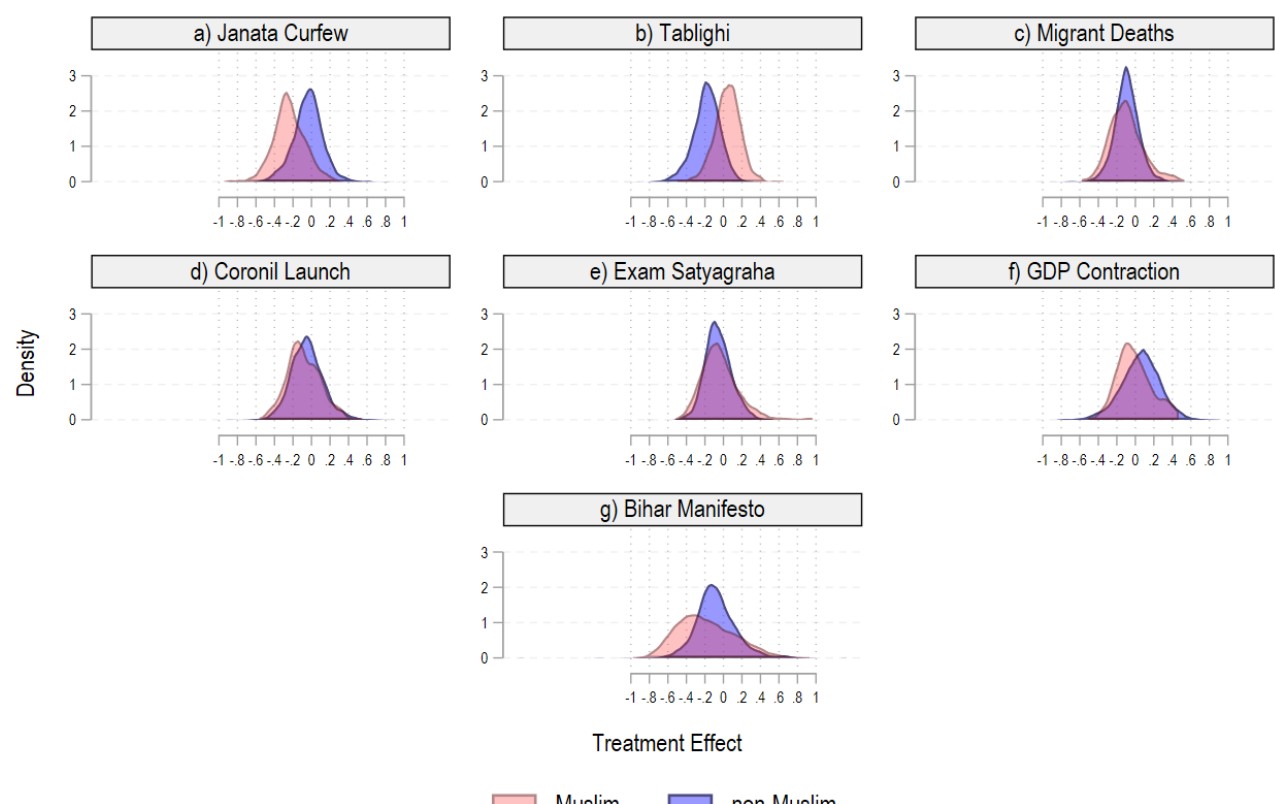

Figure 7: Effect of Interaction on GCS across Muslims and Non-Muslims

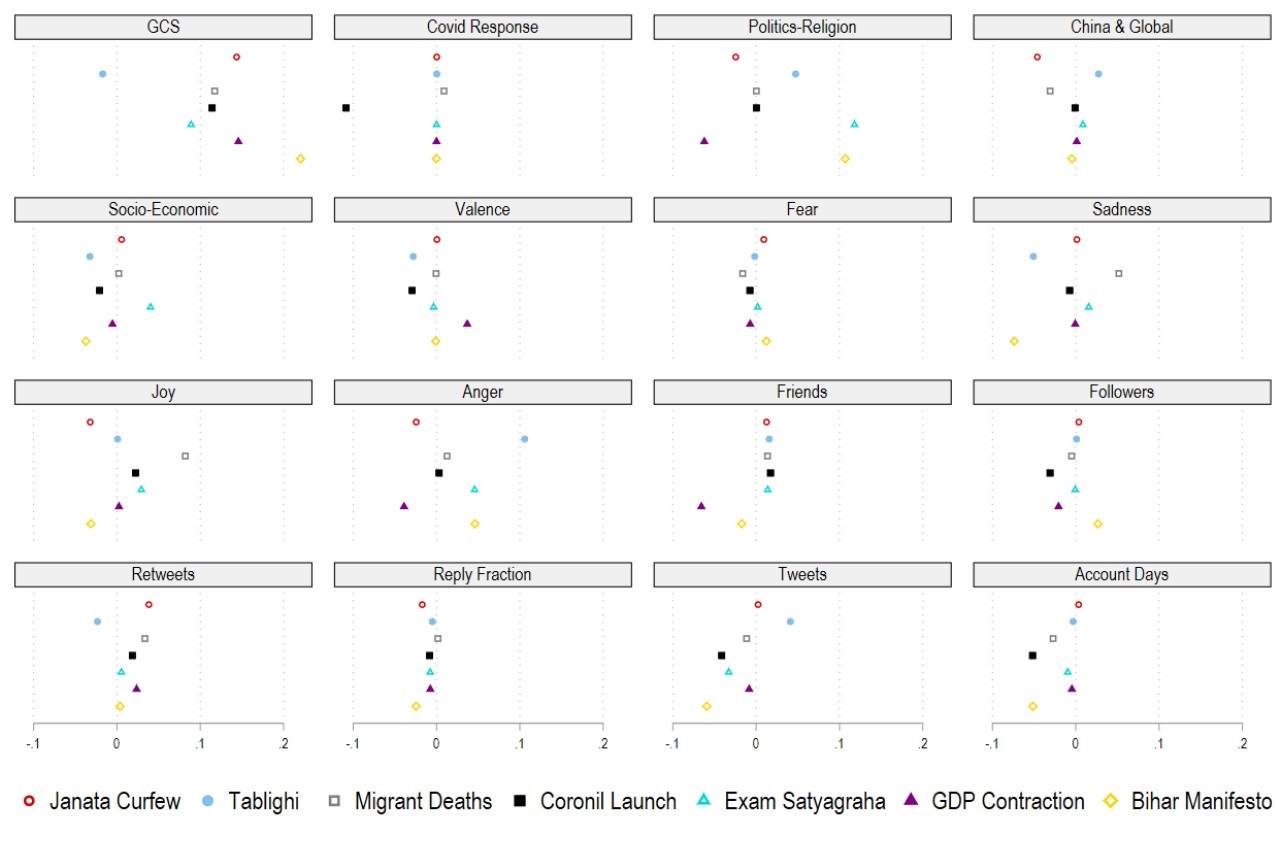

**Figure 8: Coefficient Plot of Covariates when Treatment Effect is regressed on them.**

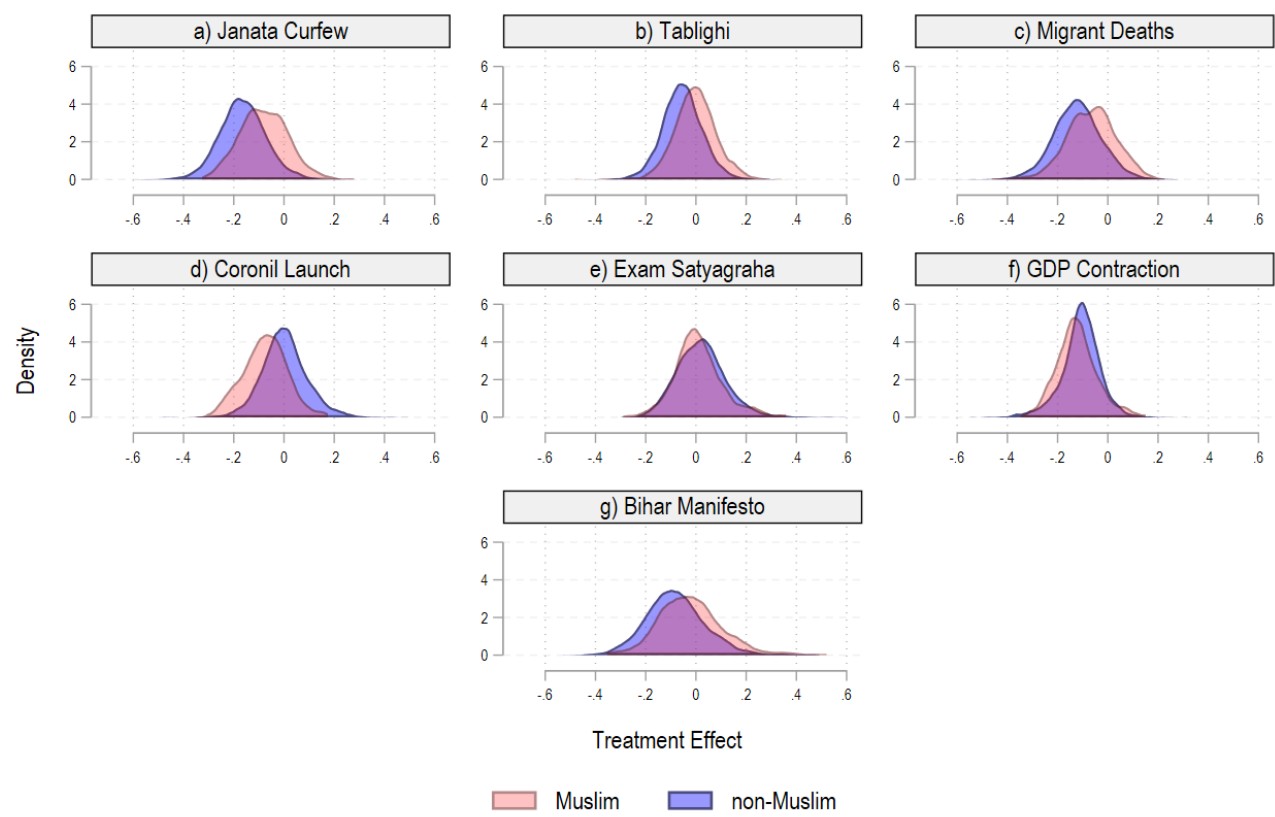

**Figure 9: Effect of Interaction on valence across Muslims and Non-Muslims**

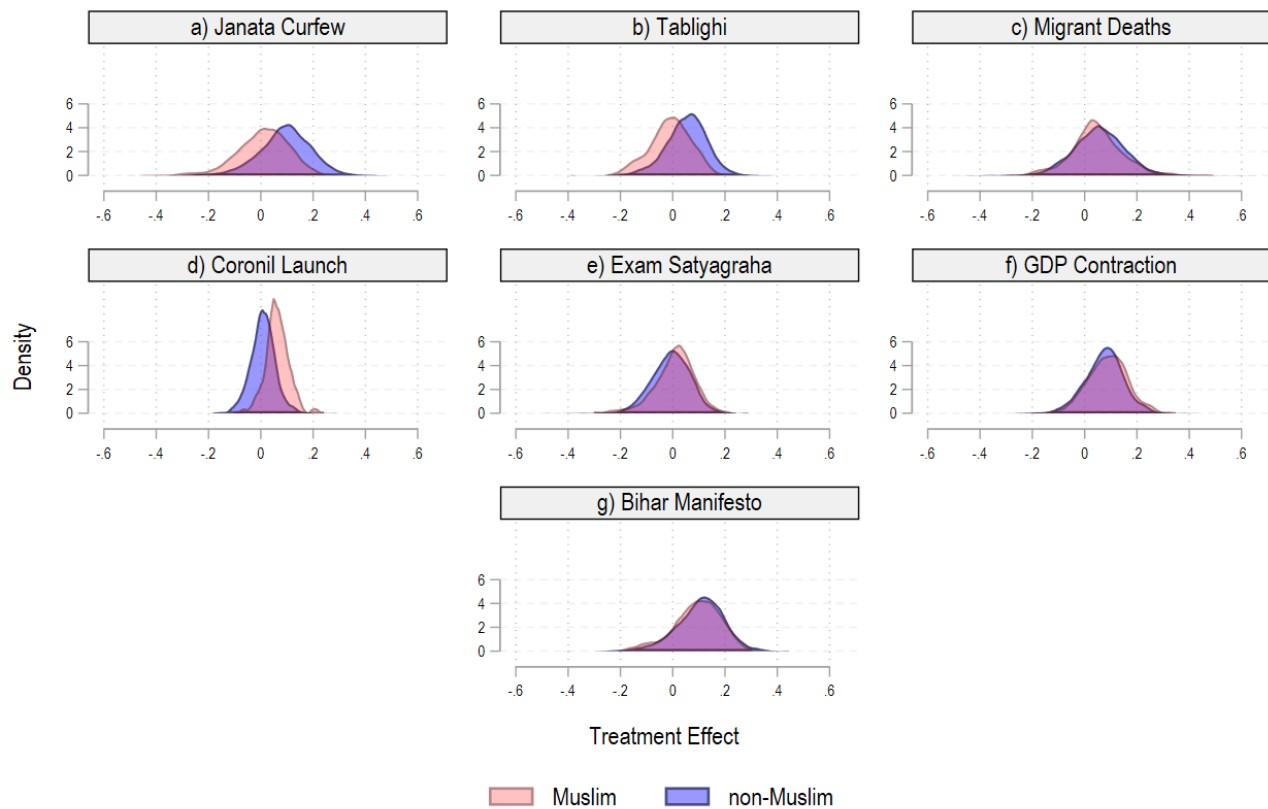

**Figure 10: Effect of Interaction on anger across Muslims and Non-Muslims**

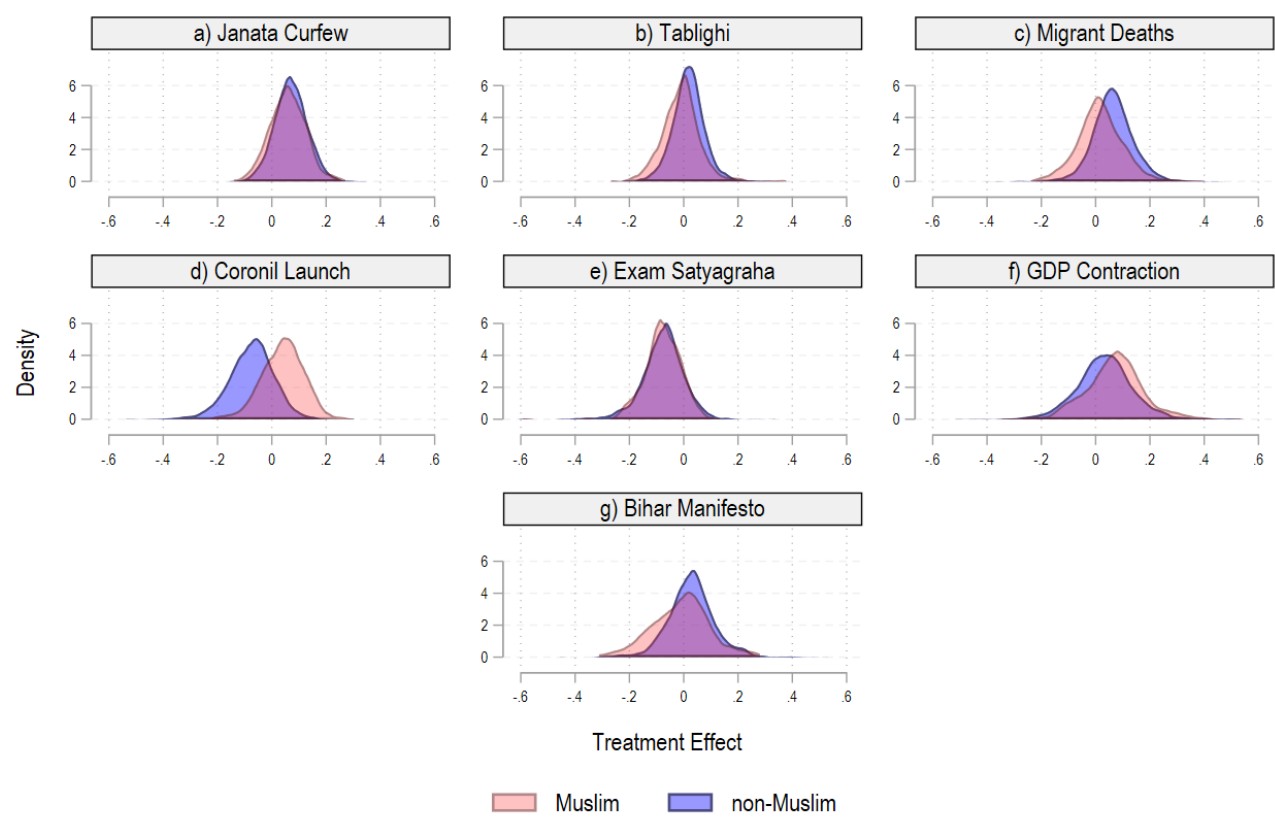

**Figure 11: Effect of Interaction on fear across Muslims and Non-Muslims**

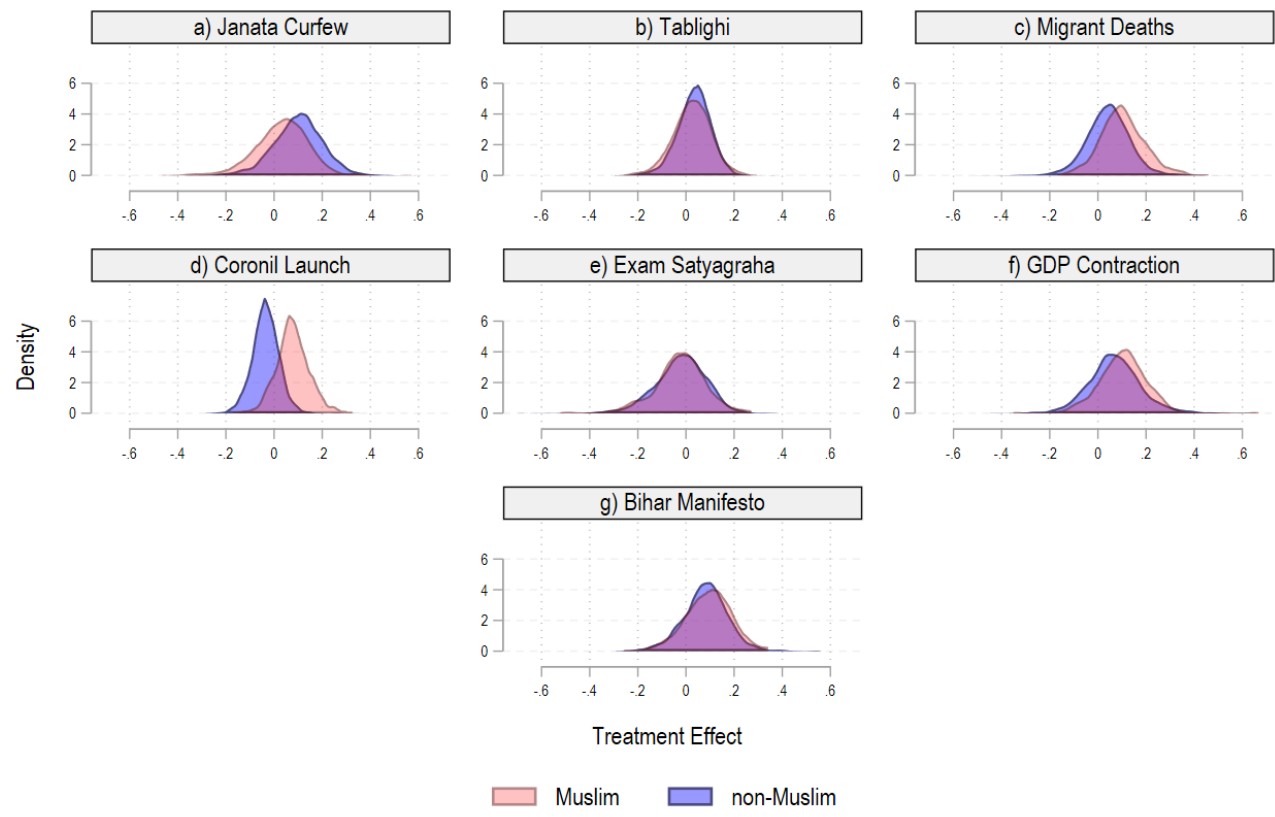

**Figure 12: Effect of Interaction on sadness across Muslims and Non-Muslims**

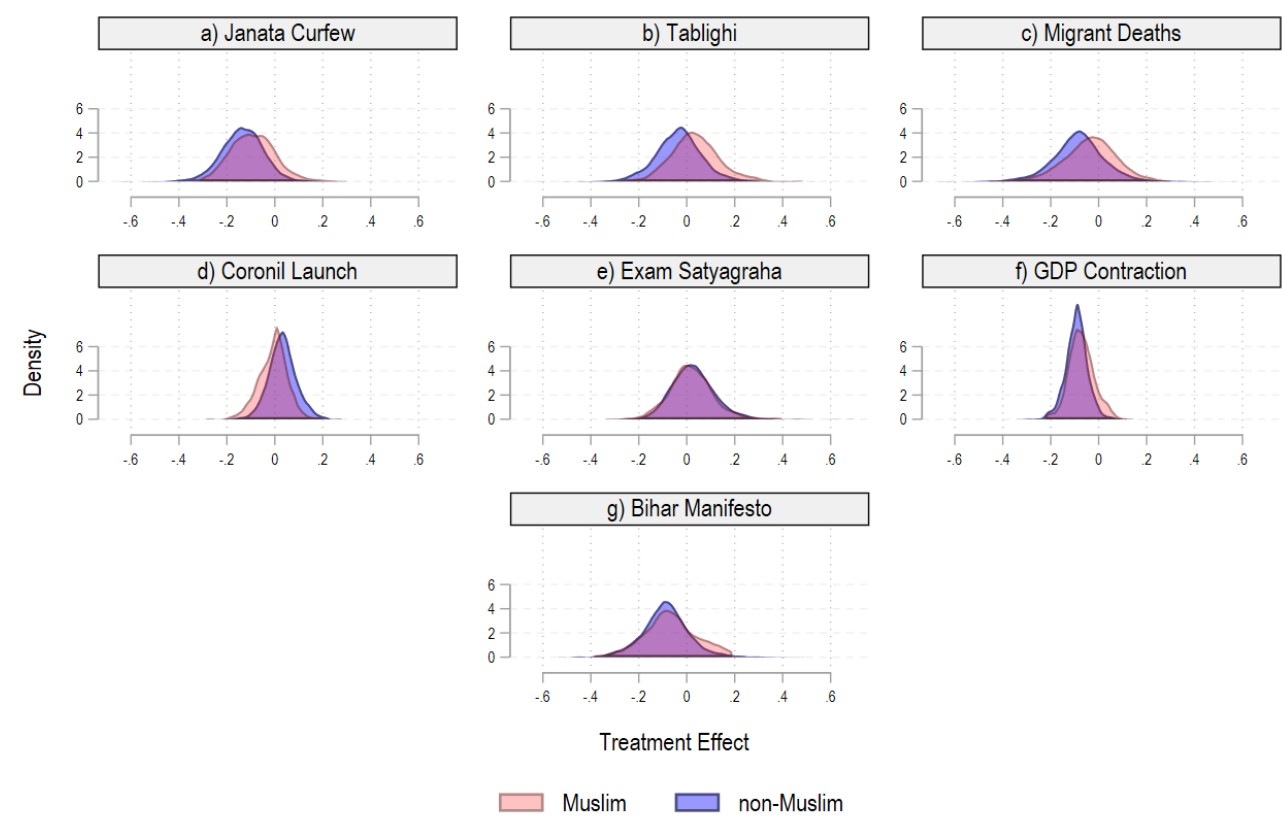

Figure 13: Effect of Interaction on joy across Muslims and Non-Muslims

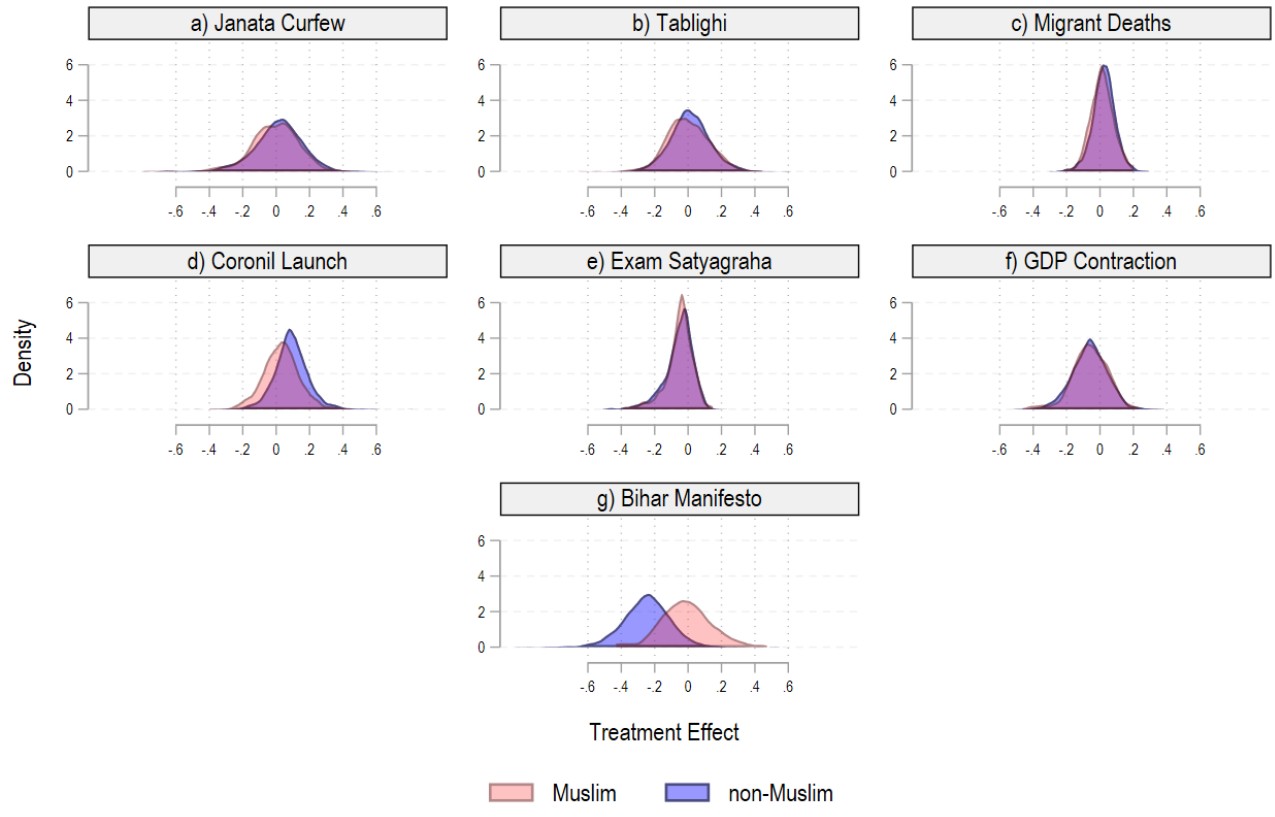

**Figure 14: Effect of Interaction on topic: COVID statistics and response across Muslims and Non-Muslims**

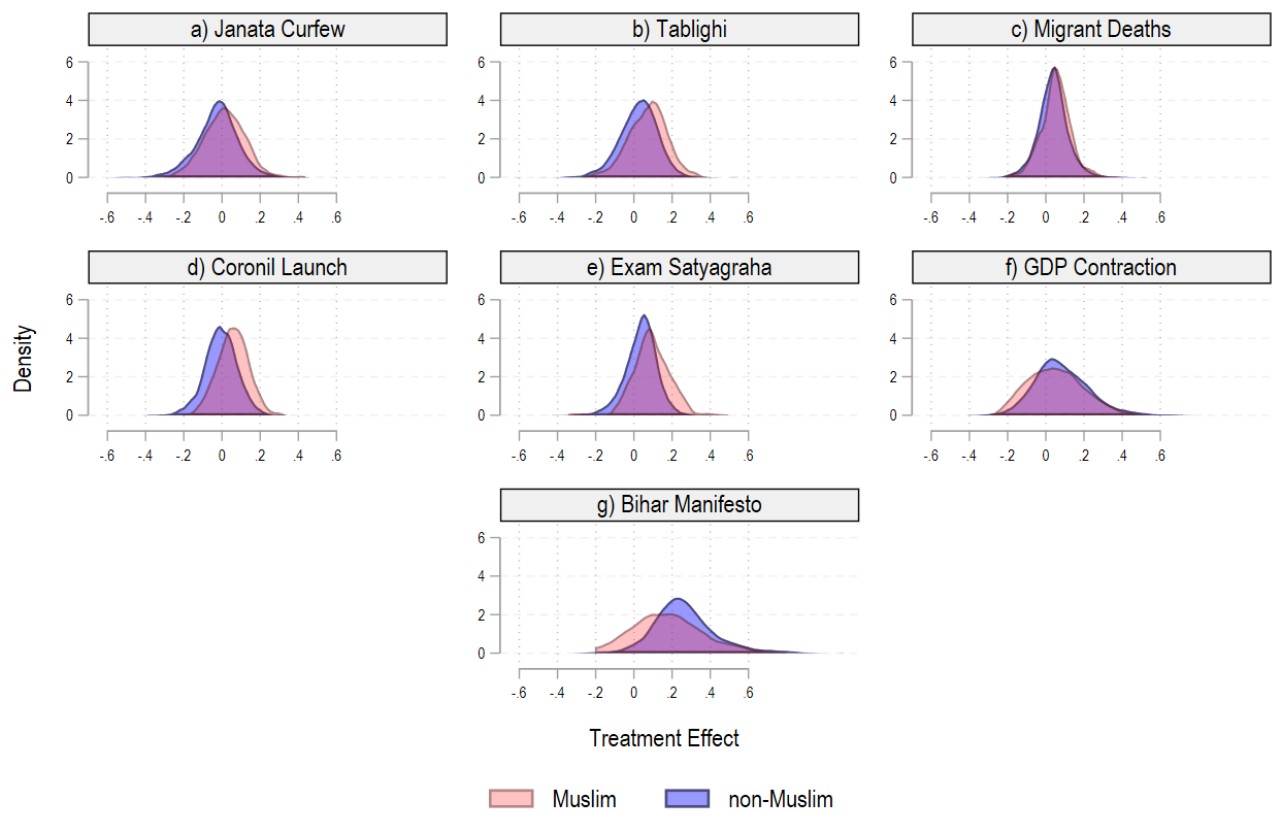

**Figure 15: Effect of Interaction on topic: politics and religion across Muslims and Non-Muslims**

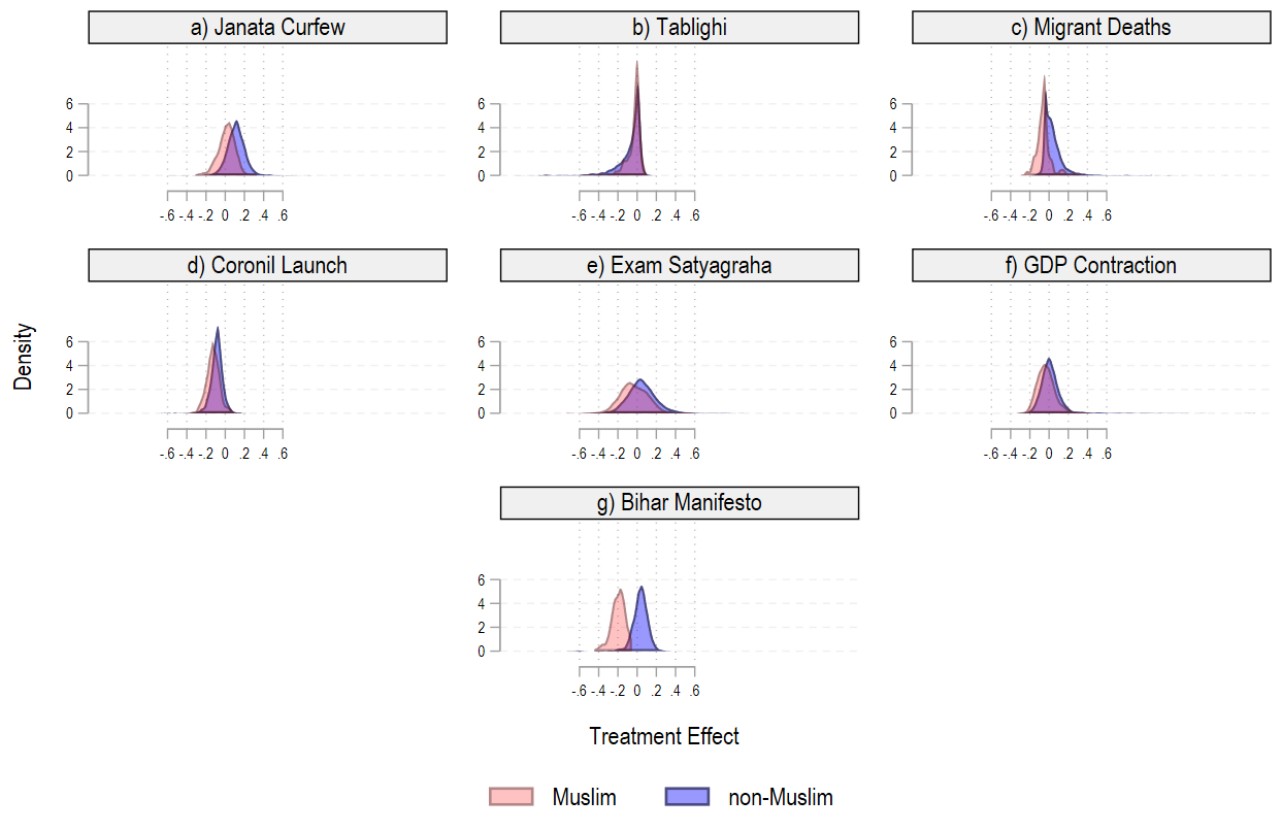

**Figure 16: Effect of Interaction on topic: China and global discourse across Muslims and Non-Muslims**

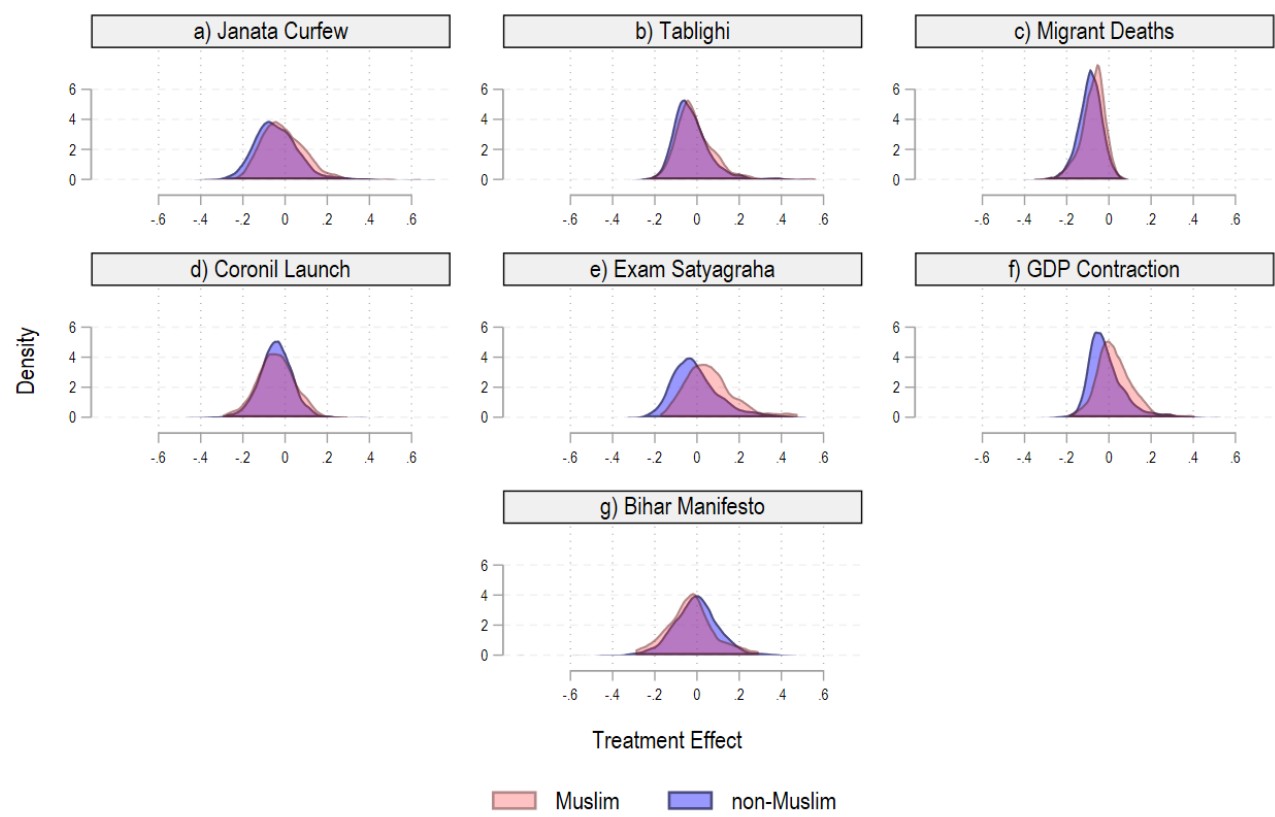

Figure 17: Effect of Interaction on topic: socio-economic issues across Muslims and Non-Muslims

