# OpenReview forum: "Bridging or Breaking: Impact of Intergroup Interactions on Religious Polarization"
_ACM.org/TheWebConf/2024/Conference — TheWebConf24_

### Official Review · Reviewer_TwFz · 2023-11-16

**Novelty:** 4
**Technical Quality:** 5

**Review:**

**Strong points**

S1. The study is timely and relevant, analyzing nearly 700,000 users offers robust insights into the dynamics of religious polarization in the context of a major global event. The paper under investigation in the paper is both valuable and of practical significance.

S2. In terms of experimental results, the proposed technical method in the paper demonstrates a noticeable improvement in quality when compared to the basic bag-of-words-based approach. The use of contextualized embeddings to measure group conformity and polarization is innovative and provides a more nuanced understanding of polarization on social media.

S3. The research uncovers important dynamics, such as the varying impacts of intergroup interactions across different types of events and the role of group conformity.

**Weak points**

W1. The reliance on Twitter data and the methods for inferring religious identity may introduce biases that could affect the results. In other words, Section 3.1 introduces a correlation between Twitter usernames, real names, and religious affiliations, which is a relatively novel proposition. However, the paper lacks sufficient evidence to substantiate its correctness.

W2. Given that the paper only investigates the interactions among Indian Twitter users during the specific context of the COVID-19 pandemic, concerns arise regarding the generalizability of the proposed method. Moreover, relying solely on this single piece of evidence may seem insufficient to support the viewpoints held by the paper.

W3. The advanced methodologies used may be difficult for readers without a background in data science or computational social science to fully comprehend.

W4. Regarding the formula $\hat{\rho}^t_{-i,d}=\sum_{j \in g_i-\{i\}}\hat{q}^t_j/(\sum_{g_i-\{i\}}\hat{q}^t_j+\sum_{\tilde{g}_i}\hat{q}^t_j)$ in Section 3.2.1, please verify its correctness.

**Questions:**

1. How does the study address potential biases in inferring religious identity from Twitter usernames?

2. Could the methodologies developed in this study be applied to other social media platforms or contexts outside of India?

3. How might the findings inform strategies for reducing religious polarization on social media platforms?

**Reviewer Confidence:**

2: The reviewer is willing to defend the evaluation, but it is likely that the reviewer did not understand parts of the paper

**Scope:**

3: The work is somewhat relevant to the Web and to the track, and is of narrow interest to a sub-community

---

### Official Review · Reviewer_xTX3 · 2023-11-20

**Novelty:** 2
**Technical Quality:** 3

**Review:**

**Summary**

This paper investigates the effects of replying to users of another religion (Muslim vs. non-muslim) on “group conformity”—a measure of how similar are a user’s tweets to their own group as opposed to tweets by the other group—in the context of COVID-19 discourse on Twitter among Indian users. The study finds that out-group replies reduce polarization but that the reduction is smaller for users who already exhibit high conformity with their group. The study also finds the effects of the intergroup interactions vary depending on the context related to the event that prompted the interaction.

**Strengths**

- Well-written and easy to follow
- Interesting research questions
- I appreciate the variety of methods used in the paper, from nested CV to the T-learned and the Oaxaca-Blinder estimator. Unfortunately, the results were also misinterpreted in important ways, as I describe below.

**Weaknesses**
- My main issue with the paper is that it misleading uses causal language throughout the manuscript to describe correlational analysis. The variable defined as “treatment” (“interact”, a binary indicator that equals 1 if a user replies to someone outside their group at least once, and 0 otherwise) is not exogenous or randomly assigned. There are numerous confounders that may lead to a reply and reduced group conformity score. For example, for a user to encounter outgroup tweets in the first place, they must have expressed some interest in the other group (e.g., followed users from the other group, interacted with outgroup tweets in the past, or searched for that content) which may also influence their likelihood to conform with their group. The application of methods that are typically used to study causal effects (such as T-Learner) does not make the estimates causal if the treatment is not randomly assigned. Note that the paper referenced next to the T-learner mention (Künzel et al., ref [25]) applies this method to data from two field experiments.

- The claimed contribution of proposing the Group Conformity Score is very small in my opinion. It is a straightforward extension of the method proposed by Gentzkow et al. to contextual embeddings. In fact, while the use of contextual embeddings improves some aspects of the score, like handling synonyms, it loses the nice interpretation of the original score that “expected posterior probability that an observer with a neutral prior would assign to a tweeter’s true party after observing a single token drawn at random from the tweets produced by the tweeter” (Demszky et al.)

- The description of how the events of interest were selected is a bit vague (“we scan the news reports”). Given that the rest of the analysis focuses on these events it would be useful if the authors could provide more details on how they ended up with this list and whether this approach was principled.

**Questions:**

Please see the comments above.

**Ethics Review Description:**

No issues identified.

**Reviewer Confidence:**

4: The reviewer is certain that the evaluation is correct and very familiar with the relevant literature

**Scope:**

3: The work is somewhat relevant to the Web and to the track, and is of narrow interest to a sub-community

---

### Official Review · Reviewer_qu5U · 2023-11-22

**Novelty:** 5
**Technical Quality:** 4

**Review:**

Positive aspects of the study:
    • The theme of the study is relevant
    • Interesting idea
Negative aspects of the study:
    • There is a lack of clarity in several parts of the text
    • Important information regarding methods is missing
    • Conclusions could not be appropriately assessed.

The authors could have posed questions in the introduction instead of raising them from section 3.

It is unclear why the authors only use 3% of eligible voters' names for constructing part names dictionary. And what is the rationale behind this approach? What is the performance?

Missed more information on the groups identified.

The procedure for identifying topics is not clear. Also, the reader wonders why the authors did not use other standard methods for this task.

What events are and how they are defined is not clear in section 3.4.

The implementation details provided in section 3.4.4 need to be clarified.

Important details regarding results in section 4.1 should be provided. For instance, the authors mention High GCS non-muslim, but the information is not evident. Also, a better discussion regarding the scores shown should be made.

There are methods mixed with results.

Due to the lack of clarity in some of the methods, it was not possible to assess properly all the conclusions made.

Other minor problems:

Problems in the text, such as "...issue, We…"… "In We then norm..."

"gmean and Youden statistics" → please let these statistics names more clear and more formalized.

**Questions:**

How do you compute the correlation mentioned in section 4.1? What data was used? Also, what is the p-value?

The reader needs to be made aware of the performance of the model to perform religious inference using names. Did you make an experiment on this? What are the results?

**Ethics Review Description:**

Nothing identified

**Reviewer Confidence:**

4: The reviewer is certain that the evaluation is correct and very familiar with the relevant literature

**Scope:**

4: The work is relevant to the Web and to the track, and is of broad interest to the community

---

### Official Review · Reviewer_FMZB · 2023-11-22

**Novelty:** 6
**Technical Quality:** 4

**Review:**

The paper explores how exposure to diverse viewpoints can affect polarization, especially around important events. By using the inferred religious beliefs of Indian Twitter users, the study looks at discussions on events during the COVID-19 pandemic in 2020. For these users, intergroup communication reduces (a specific measure of) polarization. However, for communal events, intergroup interactions can instead backfire and increase polarization.

I found the main research question of the paper very interesting. The paper presents a nice mix of qualitative and quantitative analysis, paired with a sophisticated statistical analysis framework. The conclusions are also of broad interest and refreshingly nuanced.

The paper has also several shortcomings.
- First, the paper does not provide any validation or theoretical justification for the new definition of polarization used in the paper. The literature already has a wide array of definitions, and it is not clear why this particular one should be used here, or how it differs from the existing ones.
- Second, there are a few technical points to address. The geolocation is not validated in any way, the analysis ignores the directionality of the interactions, no justification is provided for the choice of covariates for adjustment (here there should be a causal argument).
- Third, the literature referenced in the paper is quite slim, e.g., it ignores the vast literature on affective polarization from the CS point of view (a few papers published in the last edition of this conference).
- One additional problem is that the religion of the user is inferred from their name (and implicitly from their ethnicity). This technique is not validated for accuracy, for instance, the paper cites an example of misclassification. More importantly, it seems problematic from an ethical standpoint to infer religion from names. This issue should at least be discussed thoroughly.

A few minor points:
- The paper never defines what "linguistic polarization" is exactly.
- A number of users that do not match a name dictionary are filtered out, the paper should report the number of users filtered this way, and analyze possible biases introduced by this process.
- Is the GCS^{BOW} the cosine similarity of the two vectors?
- The rationale for the formula on line 325 is unclear. What is the co-domain of this function? What are its characteristics? While the intuition is clear, there are numerous possible operationalizations and it is unclear why this one was chosen.
- GCW^{BOW} is a similarity while GCS is a distance.
- Transformers have been reported to be lacking in stance detection tasks (e.g., https://link.springer.com/article/10.1007/s13218-021-00714-w). The GCS metric should be thoroughly evaluated before using it to draw conclusions. For instance, topical/lexical distance might be completely unrelated to polarization (the paper cites the example of Eid where Muslims all use the same forms of greetings and well wishes).
-

**Questions:**

Q1) How can you justify and validate the definition of the polarization measure?

Q2) What are the ethical issues of inferring religion from names?

Q3) How does the paper relate to the vast literature in CS about affective polarization?

**Ethics Review Description:**

Inference of religious beliefs from names and implied ethnicity

**Ethics Review Flag:**

Yes

**Reviewer Confidence:**

4: The reviewer is certain that the evaluation is correct and very familiar with the relevant literature

**Scope:**

4: The work is relevant to the Web and to the track, and is of broad interest to the community

---

### Official Review · Reviewer_P6Ys · 2023-11-26

**Novelty:** 6
**Technical Quality:** 7

**Review:**

In this work, authors study the effect of intergroup interactions on polarization, using a data set of COVID-related tweets from India, and focusing on the Muslim and non-muslim groups, and how polarized they are. They find that intergroup interaction between this group reduces polarization, in the sense that they make the two groups more similar in terms of words and concepts they employ.

I believe this work is in general of very high quality, from the choice of the research question to the methods employed, to the writing and presentation of results. I will list below some strong points and some areas for improvement.

### Strong points

S1. The research question posed by the authors is extremely interesting. They are able to develop an effective method to operationalize it in practice, using an existing data set in a novel way, and translating a very theoretical social science RQ into a concrete hypothesis (lines 134-136). The context of this data set also makes conclusions particularly interpretable and close to real-world implications.

S2. Methods are effective, and in particular, the novel measure of polarization developed by authors (using contextualized embeddings) is an interesting innovation. The choice of a metric of polarization that measures the content produced by individuals instead of a network-based one seems particularly effective in this context to measure the effect of cross-pollination between social groups, abstracting away the problem of negative and positive interactions that might affect network-based polarization metrics here.

S3. The problem is well-contextualized in literature: the work presents in the Introduction the related works both in terms of cross-partisan interactions and in polarization (even if some works from the past Web conferences could maybe be added).

### Weak points

W1. Authors could improve the discussion of their results, making more explicit what they think are the implications of their findings. I believe that while the work is very solid, it would be interesting to add some speculative thoughts about the causal mechanism that authors hypothesize to explain their results. I.e., in which way intergroup interactions are having the measured effect in a way that is coherent with the observations (that is, differentiating based on the 'extremeness' of individuals). Also, authors could expand of the broader significance of this result. This can be fit in the Discussion, just reducing the size of Figure 4.

W2. In Section 3.1, the authors equiparate non-muslim users in the data set to Hindus. This assumption should be better justified, especially to ensure the interpretability of the results: authors should at least provide some rough estimate of how far is this assumption from reality.

W3. Some parts of the method section could be improved in presentation: in particular, in Section 3.2 while introducing their novel metric the role of averaging the embeddings at the user-day level is not very clear. I understand the meaning in the previous metric, but its effect on embeddings seems obscure: what is the output exactly? How can it be interpreted? As relative frequencies?


### Typos

- line 203, "tcharacter"
- line 264, capital W

**Questions:**

1. Is the all-mpnet-base-v2 model employed in this work openly available? Does it ensure reproducibility?

**Ethics Review Description:**

-

**Reviewer Confidence:**

3: The reviewer is confident but not certain that the evaluation is correct

**Scope:**

4: The work is relevant to the Web and to the track, and is of broad interest to the community

---

### Decision · Program_Chairs · 2024-01-22

**Decision:**

Accept

**Comment:**

The paper investigates the impact of intergroup interactions on polarization using a dataset of COVID-related tweets from India, focusing on Muslim and non-Muslim groups. Reviewers thought it had a solid foundation and addressed important questions related to polarization in online discussions. While some concerns were raised by the reviewers, the authors' responses indicate a willingness to address these issues in the revised version. Clarifying assumptions, discussing limitations, and providing additional context can enhance the paper's quality and contribute to its overall impact.

 The paper is generally of high quality, with well-defined research questions and effective methods. The authors were responsive to reviewer feedback and willing to address concerns. The paper is well-written and clear in its presentation, as acknowledged by reviewers. Authors use contextual embeddings in measuring polarization, which adds an original dimension to the study. The paper addresses important questions related to polarization in online discussions, which is a significant and timely topic.

 Pros:

 - Interesting and relevant research questions.
 - Application of existing metrics to different data types, adding to the originality of the work.

 Cons:

 - Somewhat limited novelty.
 - Ethical concerns related to inferring religion from names, which should be addressed in the paper.
 - The need for additional clarifications and information in various sections of the paper, which the authors have committed to providing in the revised version.